Letter

# Exome sequencing identifies rare damaging variants in *ATP8B4* and *ABCA1* as risk factors for Alzheimer's disease

Alzheimer's disease (AD), the leading cause of dementia, has an estimated heritability of approximately 70%[1]. The genetic component of AD has been mainly assessed using genome-wide association studies, which do not capture the risk contributed by rare variants[2]. Here, we compared the gene-based burden of rare damaging variants in exome sequencing data from 32,558 individuals−16,036 AD cases and 16,522 controls. Next to variants in *TREM2*, *SORL1* and *ABCA7*, we observed a significant association of rare, predicted damaging variants in *ATP8B4* and *ABCA1* with AD risk, and a suggestive signal in *ADAM10*. Additionally, the rare-variant burden in *RIN3, CLU, ZCWPW1* and *ACE* highlighted these genes as potential drivers of respective AD-genome-wide association study loci. Variants associated with the strongest effect on AD risk, in particular loss-of-function variants, are enriched in early-onset AD cases. Our results provide additional evidence for a major role for amyloid-β precursor protein processing, amyloid-β aggregation, lipid metabolism and microglial function in AD.

Beyond autosomal-dominant early-onset AD (<1% of all AD cases, onset at ≤65 years), the common complex form of AD has an estimated heritability of approximately 70%[1]. Using genome-wide association studies (GWAS), 75 mostly common genetic risk factors/loci have been associated with AD risk in populations with European ancestry; however, individually these common variants have low effect sizes[2]. Using DNA sequencing strategies, rare (allele frequency <1%) damaging missense or loss-of-function (LOF) variants in the *TREM2*, *SORL1* and *ABCA7* genes were identified to also contribute to the heritability of AD, with substantially higher effect sizes than individual GWAS hits[3–8]. To detect additional genes for which rare variants are associated with AD risk, it is necessary to compare genetic sequencing data from thousands of AD cases and controls. In a large collaborative effort, we harmonized sequencing data of studies from Europe and the USA and applied a multistage gene burden analysis (Fig. 1a) (for sample descriptions, see Supplementary Table 1 and Extended Data Figs. 1 and 2). We observed site-specific technical biases, since data were generated at multiple centers, using heterogeneous methods (Supplementary Table 2). To account for these batch effects, we designed and applied comprehensive quality control (QC) procedures (Methods and Supplementary Tables 3–5).

After sample QC, we first compared gene-based rare-variant burdens between 12,652 AD cases, consisting of 4,060 early-onset AD cases (EOAD, age at onset ≤65 years) and 8,592 late-onset AD cases (LOAD, age at onset >65 years) and 8,693 controls (stage 1 analysis; Supplementary Table 3). We detected 7,543,193 variants after sample and variant QC and annotated LOF variants with LOFTEE and missense variants with the Rare Exome Variant Ensemble Learner (REVEL) score and selected variants with a minor allele frequency (MAF) < 1% (Supplementary Table 4). We defined 4 deleteriousness thresholds by incrementally including variants with lower levels of predicted deleteriousness: LOF ($n$ = 57,543), LOF + REVEL ≥ 75 ($n$ = 111,755), LOF + REVEL ≥ 50 ($n$ = 211,665) and LOF + REVEL ≥ 25 ($n$ = 409,733), respectively. Of the 19,822 autosomal protein-coding genes, we analyzed the 13,222 genes that had a cumulative minor allele count (cMAC) ≥ 10 for the lowest deleterious threshold LOF + REVEL ≥ 25 (Methods); 9,168 genes for the LOF + REVEL ≥ 50 threshold, 5,694 for the LOF + REVEL ≥ 75 threshold and 3,120 genes for the LOF-only threshold (Fig. 1b). For these different deleteriousness thresholds, this analysis has an estimated power of 41, 22, 11 and 4%, respectively to attain a signal with $P$ < 1 × 10$^{-6}$ in stage 1, assuming that for a gene, the differential variant burden between cases and controls is associated with an odds ratio (OR) of 10.0 in EOAD and 3.33 in LOAD

✉e-mail: h.holstege@amsterdamumc.nl; m.hulsman1@amsterdamumc.nl; gaelnicolas@hotmail.com; jean-charles.lambert@pasteur-lille.fr

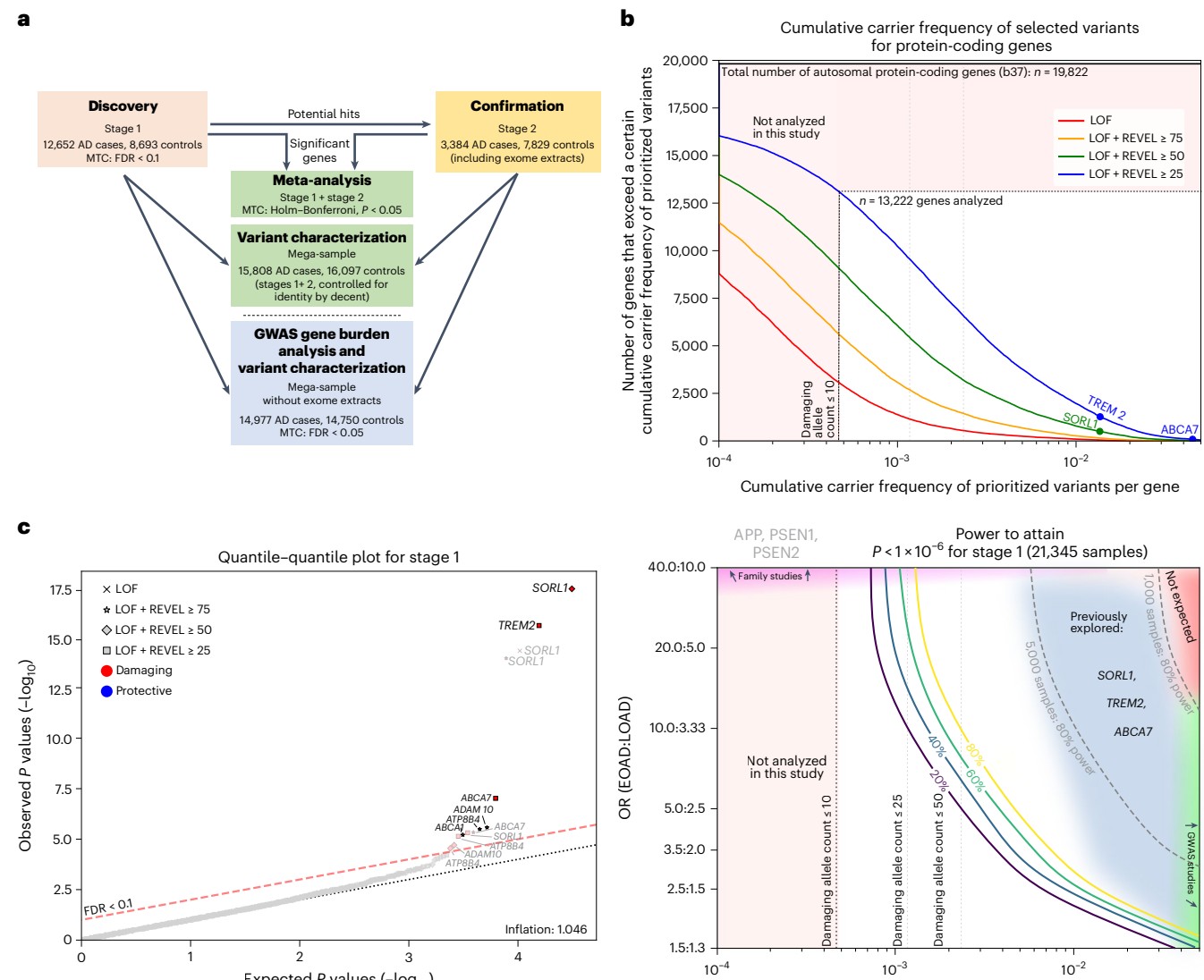

**Fig. 1 | Study setup and power. a**, Schematic of the study setup. The AD association of genes identified in stage 1 was confirmed in stage 2 and significance was determined by meta-analysis. Variant characteristics were investigated in a merged mega-sample rather than the meta-sample, allowing more accurate variant effect size estimates for variant categories/age-at-onset bins. The mega-sample (without exome extracts) was also used for the GWAS gene burden analysis. MTC, multiple testing correction. **b**, Top, number of genes (*y* axis) with at least a certain cumulative carrier frequency of prioritized variants (*x* axis), prioritized according to different deleteriousness thresholds. White box, genes with a cMAC ≥ 10 (cumulative minor allele count of ≥10 prioritized alleles identified across the 12,652 cases and 8,693 controls in the stage 1 sample) were considered to have sufficient carrier frequency to allow burden analysis. The *SORL1*, *TREM2* and *ABCA7* genes are indicated, revealing that carriers of rare damaging variants in these genes are relatively common, allowing identification in smaller sample sizes[3-7]. Bottom, power analysis for stage 1, to attain a *P* < 1 × 10⁻⁶, at the same scale as the top figure. For comparison, we indicate 80% power thresholds for sample sizes of 1,000 and 5,000 individuals (subsampled

from stage 1). Cumulative carrier frequency and estimated effect size ranges are indicated for common variants identified to associate with AD by GWAS (green), rare-variant burdens in *SORL1*, *TREM2* and *ABCA7* identified using sequencing studies[3-7] (grey/blue), and for rare variants observed in autosomal dominant AD (magenta). Common variants with high effect sizes (red) are not expected to exist. Genes with cMAC < 10 were not analyzed (pink). Power calculations show that aggregating more cases and controls might allow for the identification of rare-variants that have a large effect on AD but for which only few carriers are observed, or for variants that have a modest/average effect on AD, for which many carriers are observed (power calculations shown in Supplementary Table 6). **c**, Quantile–quantile plot of *P* values determined in the stage 1 discovery analysis based on an ordinal logistic burden test. For each of 13,222 genes, we tested the burden of variants adhering to four variant deleteriousness thresholds, conditional on having a cMAC ≥ 10 (*n* = 31,204 tests). Threshold for multiple testing correction: FDR < 0.1, *P* value inflation, 1.046. Gene names in black indicate the deleteriousness threshold of the most significant burden test in that gene.

(Supplementary Table 6). Therefore, this analysis has only the power to discover genes for which either the differential variant burden is associated with a large effect size, and/or genes for which large numbers of damaging variant carriers are observed (Fig. 1b). Using ordinal logistic regression, 31,204 burden tests were performed across 13,222 genes in stage 1 (single genes were tested with up to 4 thresholds). Statistical inflation of test results was negligible (*λ* = 1.046; Fig. 1c). Of

all the burden tests performed, 13 tests, covering 6 genes, indicated a differential rare-variant burden between AD cases and controls (false discovery rate (FDR) < 0.1): *SORL1*, *TREM2*, *ABCA7*, *ATP8B4*, *ADAM10* and *ABCA1* (Table 1)).

To confirm these signals, we applied an analysis model consistent with stage 1 to an independent stage 2 dataset, which after QC, consisted of 3,384 cases and 7,829 controls (Supplementary Table 3–5)

**Table 1 | Stages 1 and 2 and meta-analysis AD association statistics**

| Gene | Variant deleteriousness threshold | Stage 1 (n=21,345) | | | | Stage 2 (n=11,213) | | | Meta-analysis (n=32,558) | | | |
|---|---|---|---|---|---|---|---|---|---|---|---|---|
| | | P | FDR | No. variants/ no. carriers | Case/control OR (95% CI) | Pª | No. variants/ no. carriers | Case/control OR (95% CI) | P | Holm–Bonferroni | Case/control OR (95% CI) | P heterogenous |
| SORL1 | LOF+REVEL≥25 | $4.8×10^{-6}$ | **0.017** | 242/917 | 1.3 (1.1–1.5) | **$1.3×10^{-6}$** | 122/478 | 1.5 (1.2–1.9) | $1.5×10^{-10}$ | **$4.7×10^{-6}$** | 1.4 (1.2–1.5) | $1.6×10^{-1}$ |
| | LOF+REVEL≥50 | $4.0×10^{-18}$ | **<0.0001** | 167/290 | 2.6 (2.0–3.2) | **$1.4×10^{-9}$** | 79/137 | 2.4 (1.7–3.5) | $8.1×10^{-26}$ | **$2.5×10^{-21}$** | 2.5 (2.1–3.1) | $9.8×10^{-1}$ |
| | LOF+REVEL≥75 | $1.1×10^{-14}$ | **<0.0001** | 96/164 | 3.3 (2.4–4.6) | **$5.2×10^{-10}$** | 45/82 | 3.9 (2.3–6.6) | $1.1×10^{-22}$ | **$3.4×10^{-18}$** | 3.5 (2.7–4.6) | $4.3×10^{-1}$ |
| | LOF | $4.7×10^{-15}$ | **<0.0001** | 37/48 | 15.6 (3.7–37.3) | **$1.6×10^{-6}$** | 16/20 | 16.3 (3.8–35.0) | $3.3×10^{-18}$ | **$1.0×10^{-13}$** | 16.0 (9.5–27.0) | $9.4×10^{-1}$ |
| TREM2 | LOF+REVEL≥25 | $2.6×10^{-16}$ | **<0.0001** | 17/291 | 3.6 (2.9–4.6) | **$1.6×10^{-7}$** | 12/155 | 2.4 (1.6–3.4) | $5.2×10^{-22}$ | **$1.6×10^{-17}$** | 3.2 (2.6–3.9) | $6.5×10^{-1}$ |
| ABCA7 | LOF+REVEL≥25 | $9.5×10^{-8}$ | **0.001** | 265/959 | 1.4 (1.2–1.6) | **$9.8×10^{-8}$** | 170/502 | 1.6 (1.3–2.0) | $4.1×10^{-13}$ | **$1.3×10^{-8}$** | 1.4 (1.3–1.6) | $6.5×10^{-2}$ |
| | LOF+REVEL≥75 | $4.6×10^{-6}$ | **0.017** | 93/297 | 1.6 (1.3–2.1) | **$4.8×10^{-4}$** | 54/167 | 1.8 (1.3–2.6) | $7.3×10^{-9}$ | **$2.3×10^{-4}$** | 1.7 (1.4–2.1) | $9.1×10^{-1}$ |
| ATP8B4 | LOF+REVEL≥25 | $7.2×10^{-6}$ | **0.02** | 72/575 | 1.5 (1.3–1.8) | **$3.3×10^{-3}$** | 40/286 | 1.4 (1.0–1.8) | $9.6×10^{-9}$ | **$3.0×10^{-4}$** | 1.5 (1.3–1.7) | $9.7×10^{-1}$ |
| | LOF+REVEL≥50 | $2.8×10^{-5}$ | 0.068 | 61/521 | 1.5 (1.3–1.9) | **$1.6×10^{-2}$** | 34/265 | 1.3 (1.0–1.7) | $2.8×10^{-6}$ | **$8.7×10^{-2}$** | 1.5 (1.3–1.7) | $6.6×10^{-1}$ |
| | LOF+REVEL≥75 | $3.2×10^{-6}$ | **0.014** | 38/490 | 1.7 (1.4–2.0) | **$2.4×10^{-2}$** | 22/243 | 1.3 (1.0–1.8) | $5.7×10^{-7}$ | **$1.8×10^{-2}$** | 1.5 (1.3–1.8) | $4.2×10^{-1}$ |
| ABCA1 | LOF+REVEL≥75 | $6.1×10^{-6}$ | **0.019** | 93/280 | 1.7 (1.3–2.2) | **$6.6×10^{-3}$** | 48/159 | 1.6 (1.1–2.3) | $2.6×10^{-7}$ | **$8.0×10^{-3}$** | 1.7 (1.4–2.1) | $6.3×10^{-1}$ |
| ADAM10 | LOF+REVEL≥50 | $2.0×10^{-5}$ | **0.051** | 15/17 | 3.2 (1.3–8.1) | **$4.0×10^{-2}$** | 4/4 | 8.1 (0.6–42.6) | $2.8×10^{-5}$ | $8.7×10^{-1}$ | 3.6 (1.5–8.5) | $5.5×10^{-1}$ |
| | LOF+REVEL≥75 | $2.7×10^{-6}$ | **0.014** | 11/12 | 7.5 (1.4–46.8) | $1.5×10^{-1}$ | 3/3 | 5.6 (0.3–41.8) | $4.4×10^{-4}$ | $1.0×10^{0}$ | 7.1 (2.6–19.3) | $1.1×10^{-1}$ |

Listed in this table are the two-sided tests that were significant in stage 1, after multiple testing correction using a Benjamini–Hochberg FDR <0.1 over 31,204 tests/variant categories. The P values for the burden tests were determined using ordinal logistic regression; a case/control OR was computed for reference. ªIn stage 2, we considered only the direction of the AD association observed in stage 1 (that is, one-sided testing). The meta-analysis indicates the combined significance from stages 1 and 2 (data were combined using the fixed-effect inverse variance method); multiple testing correction for the meta-analysis was performed across all 31,204 tests using the Holm–Bonferroni correction (<0.05). Bold text indicates significant P values.

**Table 2 | GWAS-targeted analysis in a mega-dataset without exome extracts**

| Locus sentinel GWAS SNP | Gene | Variant deleteriousness threshold | Burden test (variant MAF <1%) | | | | Burden test (variant MAF <0.1%) | | | |
|---|---|---|---|---|---|---|---|---|---|---|
| | | | P | FDR | No. variants/ no. carriers | Case/control OR (95% CI) | P | No. variants/ no. carriers | Fraction of very rare variants, % | Case/control OR (95% CI) |
| ªSORL1, TREM2, ABCA7 (Table 1 and Supplementary Table 8) | | | | | | | | | | |
| SLC24A4/RIN3 rs7401792 rs12590654 | RIN3 | LOF+REVEL≥25 | $1.6×10^{-5}$ | 0.0003 | 44/622 | 1.4 (1.2–1.6) | $3.4×10^{-2}$ | 42/129 | 21 | 1.4 (1.0–2.1) |
| | | LOF+REVEL≥50 | $1.0×10^{-5}$ | 0.0002 | 23/583 | 1.4 (1.2–1.7) | $1.5×10^{-2}$ | 21/89 | 15 | 1.8 (1.2–2.8) |
| ªADAM10, ABCA1 (Table 1 and Supplementary Table 8) | | | | | | | | | | |
| PTK2B/CLU rs73223431 rs11787077 | CLU | LOF+REVEL≥25 | $5.0×10^{-4}$ | 0.005 | 24/26 | 3.6 (1.6–8.3) | **$5.0×10^{-4}$** | **24/26** | **100** | **3.6 (1.6–8.3)** |
| | | LOF+REVEL≥50 | $1.1×10^{-3}$ | 0.001 | 14/15 | 5.4 (1.6–28.6) | **$1.1×10^{-3}$** | **14/15** | **100** | **5.3 (1.6–28.6)** |
| | | LOF+REVEL≥75 | $5.0×10^{-4}$ | 0.005 | 12/12 | 9.9 (1.6–44.0) | **$5.0×10^{-4}$** | **12/12** | **100** | **9.8 (1.6–44.0)** |
| | | LOF | $2.6×10^{-3}$ | 0.02 | 10/10 | 7.3 (1.9–27.2) | **$2.6×10^{-3}$** | **10/10** | **100** | **7.3 (1.9–27.2)** |
| SPDYE3 rs7384878 | ZCWPW1 | LOF+REVEL≥25 | $6.1×10^{-3}$ | 0.042 | 22/77 | 1.8 (1.2–2.9) | $5.0×10^{-3}$ | 21/76 | 99 | 1.8 (1.2–2.9) |
| | | LOF+REVEL≥50 | $3.1×10^{-3}$ | 0.022 | 16/70 | 1.9 (1.2–3.1) | **$3.1×10^{-3}$** | **16/70** | **100** | **1.9 (1.2–3.1)** |
| | | LOF+REVEL≥75 | $1.1×10^{-3}$ | 0.001 | 11/15 | 5.0 (1.9–13.5) | **$1.1×10^{-3}$** | **11/15** | **100** | **5.0 (1.9–13.5)** |
| | | LOF | $7.8×10^{-4}$ | 0.008 | 11/15 | 5.0 (1.9–13.5) | **$7.8×10^{-4}$** | **11/15** | **100** | **5.0 (1.9–13.5)** |
| ACE rs4277405 | ACE | LOF+REVEL≥75 | $9.0×10^{-4}$ | 0.008 | 38/99 | 2.0 (1.3–2.9) | **$9.3×10^{-4}$** | **38/99** | **100** | **2.0 (1.3–2.9)** |

Genes in all GWAS loci were prioritized as described in the Methods (Supplementary Table 7). Listed are genes for which burden tests were significant in the mega-analysis after multiple testing correction using a Benjamini–Hochberg FDR <0.05. P values for burden tests were determined using ordinal logistic regression (two-sided tests); a case/control OR was computed for reference. ªThese genes also included the SORL1, TREM2, ABCA7, ADAM10 and ABCA1 genes, which were also identified in the rare-variant burden analysis shown in Table 1 and therefore are not shown (see Supplementary Table 8 for the full analysis). Bold text: result of burden test MAF <0.1% unchanged compared to the burden test MAF <1%.

and also with negligible P value inflation ($\lambda = 1.016$; Extended Data Fig. 3). The effect was tested in the direction observed in stage 1 (one-sided test). All genes selected in stage 1 reached P < 0.05 (Table 1, stage 2). The stage 2 effect sizes of these genes correlated with those observed in stage 1 (Pearson's r on log odds = 0.91). We then meta-analyzed stage 1 + stage 2 across the 13 tests using a fixed-effect inverse variance method and corrected for the 31,204 tests performed in stage 1 (Holm–Bonferroni) (Table 1). This confirmed the AD association of rare damaging variants in the SORL1, TREM2, ABCA7, ATP8B4 and ABCA1 genes. The association signal of the ADAM10 gene was not significant exome-wide,

presumably because prioritized variants in this gene are extremely few and rare, such that the signal can be confirmed only in larger datasets.

Strikingly, most of these genes also map to GWAS loci (SORL1, TREM2, ABCA7, ABCA1 and ADAM10). This led us to perform a focused analysis on GWAS loci, aiming to identify potential driver genes. To maximize statistical power, we merged the full exomes from the stage 1 and stage 2 samples into one mega-sample, again with negligible P value inflation ($\lambda = 1.025$; Extended Data Fig. 4). We interrogated genes that were previously prioritized to drive the AD association in the 75 loci identified in the most recent GWAS[2] (Supplementary Table 7

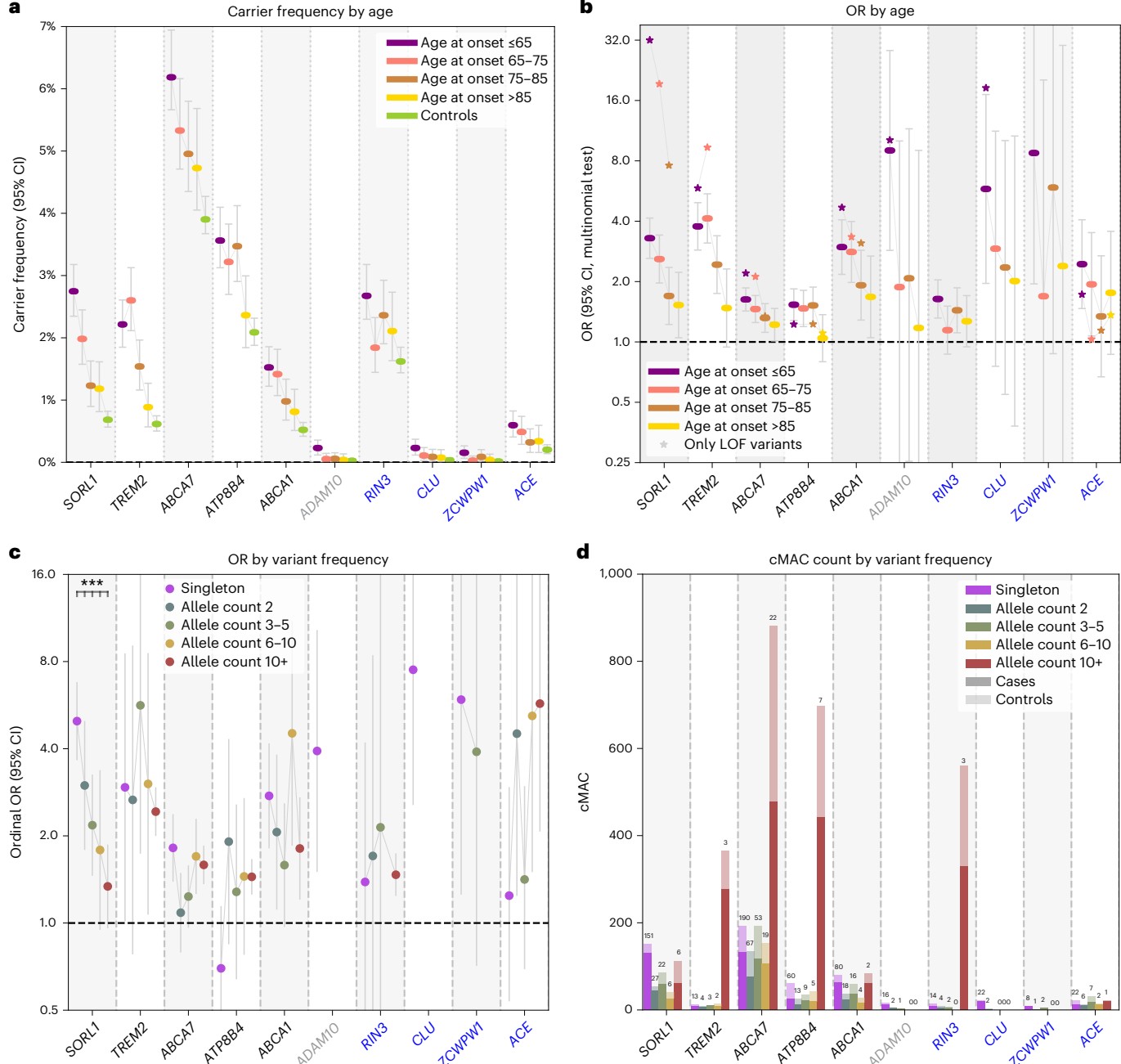

**Fig. 2 | Characterization of gene-specific variant features based on the mega-sample.** For all variant features, we considered the deleteriousness threshold that provides the most evidence for AD association in the meta-analysis. Variant features were investigated in a merged mega-sample (*n* = 31,905) instead of the meta-sample because this allows for increased accuracy for estimations of variant effect sizes for each variant category/ age-at-onset bin (Table 3, refined burden). **a**, Carrier frequency according to age at onset. A carrier carries at least one damaging variant in the considered gene. **b**, ORs according to age at onset. The effect size significantly decreased with age at onset for *SORL1*, *TREM2*, *ABCA7*, *ABCA1* and *ADAM10* (after multiple testing

correction; Supplementary Table 9). **c**, ORs according to variant frequency. The rareness of variants in *SORL1* was significantly associated with the effect size (Supplementary Table 11). **d**, cMAC by variant frequency: the stacked total number of cases (dark) and controls (light) that carry gene variants with allele frequencies as observed in the mega-sample. The numbers above the bars indicate the number of contributing variants. Whiskers: 95% CI. Genes in black: genes identified to significantly associate with AD in the meta-analysis; gray: genes not significantly associated with AD in the meta-analysis; blue: genes identified by the targeted GWAS analysis, these were not significantly associated with AD in the meta-analysis.

and Methods). In 67 genes, we observed sufficient prioritized variants (cMAC ≥ 10) to test the burden signal in at least 1 deleteriousness category (a total of 187 tests). In addition to the genes mentioned above, our analysis indicated a suggestive signal of increased AD risk in *RIN3*, *CLU*, *ZCWPW1* and *ACE* (FDR < 0.05) (Table 2 and Supplementary Table 8); these signals will have to be confirmed in a larger dataset. Nevertheless,

the AD associations in these genes persisted when focusing on the burden of only the very rare variants (MAF < 0.1%), suggesting that the rare-variant burden is not in linkage with, and thus independent from, the GWAS sentinel variant.

Together, the newly associated genes provide additional evidence for a central role for APP processing, lipid metabolism, amyloid-β (Aβ)

**Table 3 | Mega-analysis: carrier frequency, effect sizes, median age at onset and attributable fraction**

| | Mega-analysis | | Carrier frequency | | OR (95% CI) | | | Median age at onset (IQR) | Attributable fraction |
|---|---|---|---|---|---|---|---|---|---|
| | Gene | Group | No. variant/ no. carriers | EOAD/LOAD/ controls, % | Case/control | EOAD/control | LOAD/control | | EOAD/ LOAD, % |
| **Primary analysis** | *SORL1* | **LOF+REVEL≥50** | **212/418** | **2.75/1.51/0.68** | **2.5 (2.0–3.0)** | **3.3 (2.6–4.1)** | **2.0 (1.6–2.5)** | **65 (59–73)** | **1.91/0.75** |
| | | —Missense (REVEL 50–100) | 161/354 | 2.02/1.31/0.66 | 2.1 (1.7–2.5) | 2.5 (2.0–3.2) | 1.8 (1.4–2.3) | 67 (59–74) | 1.22/0.58 |
| | | —LOF | 51/68 | 0.78/0.21/0.02 | 19.8 (11.9–32.7) | 40.7 (12.5–133) | 11.3 (3.3–38.3) | 62 (56–69) | 0.76/0.19 |
| | *TREM2* | **LOF+REVEL≥25** | **26/441** | **2.27/1.90/0.75** | **2.8 (2.3–3.5)** | **3.3 (2.6–4.3)** | 2.6 (2.1–3.3) | **69 (62–75)** | **1.58/1.17** |
| | | **LOF+REVEL≥25 (refined)** | **25/404** | **2.22/1.77/0.62** | **3.1 (2.6–3.8)** | **3.8 (2.9–4.9)** | **2.8 (2.2–3.6)** | **68 (62–75)** | **1.63/1.15** |
| | | —Missense (REVEL 25–100) | 14/377 | 2.06/1.63/0.59 | 3.0 (2.5–3.8) | 3.7 (2.8–4.9) | 2.7 (2.1–3.6) | 68 (62–75) | 1.50/1.04 |
| | | —LOF | 12/66 | 0.21/0.29/0.16 | 2.1 (1.2–3.4) | 1.7 (0.8–3.5) | 2.2 (1.3–3.9) | 71 (63–76) | 0.09/0.16 |
| | | —LOF (refined) | 11/29 | 0.16/0.16/0.02 | 5.6 (2.6–12.1) | 5.8 (1.7–19) | 5.4 (1.8–16.8) | 71 (63–74) | 0.13/0.13 |
| | *ABCA7* | **LOF+REVEL≥25** | **351/1,489** | **6.18/5.04/3.90** | **1.4 (1.3–1.6)** | **1.6 (1.4–1.9)** | **1.3 (1.2–1.5)** | **69 (61–78)** | **2.40/1.29** |
| | | —Missense (REVEL 25–100) | 302/1,372 | 5.58/4.65/3.63 | 1.4 (1.3–1.6) | 1.6 (1.4–1.8) | 1.3 (1.2–1.5) | 69 (62–78) | 2.06/1.18 |
| | | —LOF | 49/119 | 0.62/0.39/0.27 | 1.7 (1.1–2.4) | 2.2 (1.4–3.5) | 1.4 (0.9–2.1) | 67 (57–74) | 0.34/0.11 |
| | *ATP8B4* | **LOF+REVEL≥25** | **94/850** | **3.56/3.08/2.09** | **1.4 (1.2–1.6)** | **1.5 (1.3–1.8)** | **1.4 (1.2–1.6)** | **70 (61–78)** | **1.24/0.84** |
| | | —Missense (REVEL 25–100) | 74/797 | 3.35/2.93/1.93 | 1.5 (1.3–1.7) | 1.6 (1.3–1.9) | 1.4 (1.2–1.7) | 70 (62–78) | 1.20/0.84 |
| | | —LOF | 20/54 | 0.21/0.16/0.16 | 1.1 (0.6–1.9) | 1.2 (0.6–2.4) | 1.0 (0.5–1.8) | 70 (59–78) | 0.03/−0.01 |
| | *ABCA1* | **LOF+REVEL≥75** | **122/442** | **1.91/1.50/1.13** | **1.6 (1.3–2.0)** | **1.9 (1.5–2.5)** | **1.5 (1.2–1.9)** | **70 (60–76)** | **0.91/0.48** |
| | | **LOF+REVEL≥75 (refined)** | **120/282** | **1.52/1.10/0.52** | **2.4 (1.9–3.1)** | **2.9 (2.2–4.0)** | **2.2 (1.6–2.9)** | **70 (59–76)** | **1.01/0.60** |
| | | —Missense (REVEL 75–100) | 95/395 | 1.63/1.32/1.05 | 1.5 (1.2–1.8) | 1.7 (1.3–2.2) | 1.4 (1.1–1.8) | 70 (61–76) | 0.68/0.37 |
| | | —Missense (REVEL 75–100 (refined)) | 93/235 | 1.24/0.92/0.44 | 2.3 (1.7–3.0) | 2.7 (1.9–3.8) | 2.1 (1.5–2.8) | 70 (59–76) | 0.78/0.48 |
| | | —LOF | 27/47 | 0.28/0.18/0.08 | 3.5 (1.9–6.4) | 4.7 (2.2–10.3) | 2.8 (1.3–6.1) | 67 (59–77) | 0.22/0.11 |
| | *ADAM10* | LOF+REVEL≥50 | 19/22 | 0.23/0.05/0.02 | 4.7 (2.0–10.8) | 9.0 (2.9–28) | 2.2 (0.5–8.2) | 63 (60–68) | 0.20/0.03 |
| **GWAS-targeted analysis** | *RIN3* | **LOF+REVEL≥50** | **23/583** | **2.67/2.10/1.62** | **1.4 (1.2–1.7)** | **1.6 (1.3–2.0)** | **1.3 (1.1–1.6)** | **70 (59–79)** | **1.04/0.46** |
| | | —Missense (REVEL 50–100) | 17/577 | 2.62/2.08/1.61 | 1.4 (1.2–1.7) | 1.6 (1.3–2.0) | 1.3 (1.1–1.6) | 70 (59–79) | 1.01/0.45 |
| | | —LOF | 6/8 | 0.06/0.03/0.01 | 2.1 (0.5–9.3) | 2.9 (0.5–18.0) | 1.7 (0.3–10.3) | 69 (57–86) | 0.04/0.01 |
| | *CLU* | **LOF+REVEL≥25** | **24/26** | **0.23/0.09/0.03** | **3.6 (1.6–8.3)** | **5.8 (2.0–17.1)** | 2.5 (0.8–7.6) | **63 (58–73)** | **0.19/0.05** |
| | | —Missense (REVEL 25–100) | 14/16 | 0.12/0.06/0.03 | 2.6 (0.9–7.5) | 3.6 (0.9–13.6) | 2.1 (0.6–8.0) | 68 (58–76) | 0.08/0.03 |
| | | —LOF | 10/10 | 0.12/0.03/0.01 | 7.3 (1.9–27.2) | 14.2 (2.9–470.4) | 3.8 (0.6–122.4) | 63 (59–68) | 0.11/0.02 |
| | *ZCWPW1* | **LOF** | **11/15** | **0.15/0.05/0.01** | **5.0 (1.9–13.5)** | **9.1 (2.0–42.0)** | 2.9 (0.8–14.7) | **63 (58–81)** | **0.14/0.03** |
| | *ACE* | **LOF+REVEL≥75** | **38/99** | **0.60/0.39/0.20** | **2.0 (1.3–2.9)** | **2.4 (1.5–4.1)** | 1.7 (1.0–2.7) | **67 (60–75)** | **0.35/0.16** |
| | | —Missense (REVEL 75–100) | 10/49 | 0.33/0.22/0.07 | 3.2 (1.7–5.7) | 3.9 (1.8–8.8) | 2.7 (1.3–5.9) | 66 (61–72) | 0.24/0.14 |
| | | —LOF | 28/50 | 0.27/0.16/0.14 | 1.4 (0.8–2.4) | 1.7 (0.9–3.4) | 1.2 (0.6–2.2) | 70 (55–76) | 0.11/0.02 |

For each gene, the AD association statistics are shown for the variant deleteriousness threshold with the most evidence for AD association in the meta-analysis (bold). For genes with sufficient carriers, signals are shown for LOF and missense variants separately (regular text). Individual variants contributing to the burden were validated in a multistage analysis (Supplementary Table 16 and Methods), which resulted in the construction of a refined burden for *TREM2* (one variant removed) and *ABCA1* (two variants removed). The attributable fraction of a gene is an estimate of the fraction of EOAD and LOAD cases in this sample that have become part of this dataset due to carrying a rare damaging variant in the respective gene (Methods). Note that several variants were excluded from this analysis (that is, due to differential missingness) that would otherwise have been included in the burden. See section 2 of the Supplementary Note for a gene-specific discussion of the variants that contribute to the association with AD and Supplementary Data for the list of variants considered in the burden analysis. Genes shown in bold: the variant burden was significantly associated with AD in the meta-analysis (Holm–Bonferroni <0.05; Table 1). *P* values for the mega-analysis are shown in Supplementary Table 15.

aggregation and neuroinflammatory processes in AD pathophysiology. Like *ABCA7*, *ATP8B4* encodes a phospholipid transporter. Rare variants in this gene have been associated with the risk of developing systemic sclerosis, an autoimmune disease[9]. In the brain, *ATP8B4* is predominantly expressed in microglia. Interestingly, GWAS indicated a potential association of *ATP8B4* with AD[2], mainly through the rare missense variant that was most recurrent in our study (G395S). Of note, the OR point estimate for *ATP8B4* LOF variants was close to 1, allowing

for the possibility that the missense variants that drive the *ATP8B4* association do not depend on a LOF effect. *ABCA1* also encodes a phospholipid transporter; it lipidates apolipoprotein E (APOE)[10] and poor ABCA1-dependent lipidation of APOE-containing lipoprotein particles increases Aβ deposition and fibrillogenesis[11]. In line with this, the rare N1800H LOF variant in *ABCA1* was previously associated with low plasma levels of APOE and evidence suggested an association with increased risk of AD and cerebrovascular disease[12].

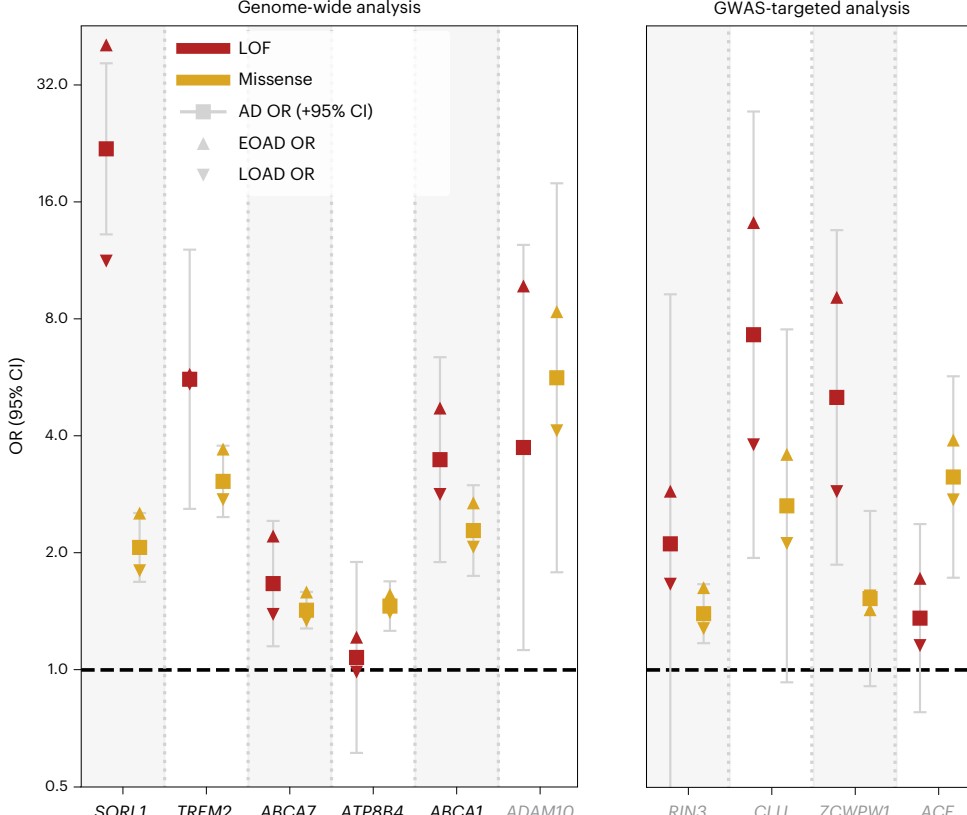

**Fig. 3 | ORs according to age at onset and variant pathogenicity.** ORs for LOF (red) and missense (yellow) variants as observed in the mega-sample (*n* = 31,905). Case/control OR (square, 95% CI), EOAD OR (triangle pointing upward), LOAD OR (triangle pointing downward). Missense variants in the considered gene appertained to the variant deleteriousness threshold that provides the most evidence for its AD association (Table 3, refined). The LOF burden effect size was significantly larger than the missense burden effect size in the *SORL1* and we observed similar trends in *ABCA7* and *ABCA1* (Supplementary Table11). Of note, for *ZCWPW1* only the burden of the LOF variants was significantly associated with AD; missense variants are shown for reference purposes (REVEL > 25). Grey: gene was not significantly associated with AD in the meta-analysis.

The α-secretase ADAM10 plays a major role in non-amyloidogenic APP metabolism[13]. Evidence for the AD association of rare variants in *ADAM10* has remained suggestive until now: two rare missense variants in *ADAM10* were reported before to incompletely segregate with LOAD in a few families[14] (these variants did not associate with AD in our study; Supplementary Data) and a nonsense variant in the *ADAM10* gene segregated with AD but in a small pedigree[15]. *RIN3* has been associated with endosomal dysfunction and APP trafficking/metabolism[16,17]. *CLU* (also known as *APOJ*) affects Aβ aggregation and clearance[18] and ACE is suggested to have a role in Aβ degradation[19]. Thus far, the role of the histone methylation reader *ZCWPW1* is unclear.

To better comprehend how these genes associate with AD, we analyzed the characteristics of rare damaging variants that contributed to the burden using the mega-sample (Fig. 2 and Table 3). For damaging variants in most genes, we observed increased carrier frequencies in younger cases and larger effect sizes were associated with an earlier age at onset (*P* = 0.0001) (Supplementary Table 9 and Extended Data Fig. 5). Yet the variants also contributed to an increased risk of LOAD (Fig. 2a,b and Table 3). The largest effect sizes were measured for LOF variants in *SORL1*, *ADAM10*, *CLU* and *ZCWPW1*; carriers of such variants had the lowest median age at onset, implying a key role for these genes in AD etiology (Table 3 and Extended Data Fig. 6). Moderate variant effect sizes were observed for LOF variants in *TREM2*, *ABCA1* and *RIN3*, while the smallest variant effects were observed in *ABCA7*, *ATP8B4* and *ACE* (Fig. 3 and Table 3).

Extremely rare variants contributed more to large effect sizes than less rare variants (*P* = 0.03; Supplementary Table 10). Indeed, for *SORL1*, the variants with the lowest variant frequencies had the largest effect sizes (Fig. 2c and Supplementary Table 11) and damaging variants in *ADAM10*, *CLU* and *ZCWPW1* were all extremely rare (Fig. 2d). Conversely, we observed that rare but recurrent variants contributed to the AD association of *TREM2*, *ABCA7*, *ATP8B4* and *RIN3* (Fig. 2d). The effect sizes of rare coding variant burdens were large compared to the effect sizes of the GWAS sentinel SNPs (Supplementary Tables 7 and 8). Up to 18% EOAD and 14% LOAD cases carried at least 1 predicted damaging variant in 1 of the 10 genes, compared to 9% of the controls (Supplementary Table 12). The fractions of EOAD cases in our sample that could be attributed to a rare variant in a specific gene ranged between 0.1 and 2.4% (approximately 2%: *SORL1*, *TREM2*, *ABCA7*; approximately 1%: *ATP8B4*, *ABCA1*, *RIN3*; and <0.5% for the remaining genes); for LOAD cases, this ranged between 0 and 1.3% (Table 3 and Extended Data Fig. 7).

We performed an age-matched sensitivity analysis to investigate possible effects from other age-related conditions, which supported a role in AD for all ten identified genes (Extended Data Fig. 8). Since *APOE* status was used as the selection criterion in several contributing datasets, burden tests were not adjusted for *APOE-ε4* dosage; in a separate analysis we observed no interaction effects between the rare-variant AD association and *APOE-ε4* dosage (Supplementary Table 13 and Methods). Also, the rare-variant burden association was not confounded by somatic mutations due to age-related clonal hematopoiesis (Supplementary Table 14).

Together, we report *ATP8B4* and *ABCA1* as new AD risk factors with exome-wide significance and we report suggestive evidence for the association of rare variants in the *ADAM10* gene with AD risk.

Furthermore, we identified *RIN3*, *CLU*, *ZCWPW1* and *ACE* as potential drivers in GWAS loci, illustrating how analyses of rare protein-modifying variants can solve this drawback of GWAS studies[20]. Larger datasets will be required to further confirm these signals. Given the association of LOF variants with increased AD risk, we suggest that the GWAS risk alleles in the respective loci might also be associated with reduced activity of the gene, which will have to be evaluated in further experiments. We observed an increased burden of rare damaging genetic variants in individuals with an earlier age at onset. Nevertheless, damaging variants (including *APOE-ε4/ε4*) were observed in only 30% of the EOAD cases (Supplementary Table 12), suggesting that additional damaging variants are yet to be discovered (Fig. 1b). Further, the effect of structural variants such as copy number variants and repetitive sequences will need to be investigated in future analyses. The associated genes strengthen our current understanding of AD pathophysiology. When treatment options become available in the future, identification of damaging variants in these genes will be of interest to clinical practice.

## Online content

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

Henne Holstege [1,2,3,4,74] ✉, Marc Hulsman [1,2,3,4,74] ✉, Camille Charbonnier[5,74], Benjamin Grenier-Boley[6], Olivier Quenez [5], Detelina Grozeva [7], Jeroen G. J. van Rooij[8,9], Rebecca Sims [7], Shahzad Ahmad [10,11], Najaf Amin [10,12], Penny J. Norsworthy [13], Oriol Dols-Icardo [14,15], Holger Hummerich[13], Amit Kawalia[16], Philippe Amouyel [6], Gary W. Beecham[17], Claudine Berr [18], Joshua C. Bis [19], Anne Boland[20], Paola Bossù [21], Femke Bouwman[2,3], Jose Bras[22,23], Dominique Campion[5], J. Nicholas Cochran [24], Antonio Daniele[25], Jean-François Dartigues[26], Stéphanie Debette [26,27], Jean-François Deleuze[20], Nicola Denning[28], Anita L. DeStefano[29,30,31], Lindsay A. Farrer [29,31,32,33], Maria Victoria Fernández [34,35,36], Nick C. Fox [37], Daniela Galimberti [38,39], Emmanuelle Genin [40], Johan J. P. Gille[41], Yann Le Guen [42], Rita Guerreiro[22,23], Jonathan L. Haines [43], Clive Holmes[44], M. Arfan Ikram [10], M. Kamran Ikram[10], Iris E. Jansen[2,3,45], Robert Kraaij [9], Marc Lathrop[46], Afina W. Lemstra[2,3], Alberto Lleó[14,15], Lauren Luckcuck[7], Marcel M. A. M. Mannens[47], Rachel Marshall[7], Eden R. Martin[17,48], Carlo Masullo[49], Richard Mayeux[50,51], Patrizia Mecocci[52], Alun Meggy[28], Merel O. Mol [8], Kevin Morgan [53], Richard M. Myers[24], Benedetta Nacmias [54,55], Adam C. Naj [56,57], Valerio Napolioni [42,58], Florence Pasquier [59], Pau Pastor [60,61], Margaret A. Pericak-Vance [17,48], Rachel Raybould [28], Richard Redon [62], Marcel J. T. Reinders [4], Anne-Claire Richard[5], Steffi G. Riedel-Heller [63], Fernando Rivadeneira [9], Stéphane Rousseau[5], Natalie S. Ryan[37], Salha Saad[7], Pascual Sanchez-Juan [15,64], Gerard D. Schellenberg[57], Philip Scheltens[2,3], Jonathan M. Schott [37], Davide Seripa[65], Sudha Seshadri [30,31,66], Daoud Sie[41], Erik A. Sistermans[41], Sandro Sorbi[54,55], Resie van Spaendonk [41], Gianfranco Spalletta [67], Niccolo' Tesi [1,2,3,4], Betty Tijms[2], André G. Uitterlinden [9], Sven J. van der Lee [1,2,3,4], Pieter Jelle Visser[2], Michael Wagner[68,69], David Wallon[70], Li-San Wang[57], Aline Zarea[70], Jordi Clarimon[14,15], John C. van Swieten[8], Michael D. Greicius[42], Jennifer S. Yokoyama [71], Carlos Cruchaga[34,35,36], John Hardy[72], Alfredo Ramirez [16,66,68,69,73], Simon Mead [13], Wiesje M. van der Flier [2,3], Cornelia M. van Duijn [10,12], Julie Williams [7], Gaël Nicolas [5,74] ✉, Céline Bellenguez [6,74] & Jean-Charles Lambert [6,74] ✉

[1]Genomics of Neurodegenerative Diseases and Aging, Human Genetics, Vrije Universiteit Amsterdam, Amsterdam UMC location VUmc, Amsterdam, the Netherlands. [2]Alzheimer Center Amsterdam, Neurology, Vrije Universiteit Amsterdam, Amsterdam UMC location VUmc, Amsterdam, the Netherlands. [3]Amsterdam Neuroscience, Neurodegeneration, Amsterdam, the Netherlands. [4]Delft Bioinformatics Lab, Delft University of Technology, Delft, the Netherlands. [5]Université Rouen Normandie, INSERM U1245 and CHU Rouen, Department of Genetics and CNRMAJ, Rouen, France. [6]Université Lille, INSERM, Centre Hospitalier Universitaire Lille, Institut Pasteur de Lille, U1167-RID-AGE facteurs de risque et déterminants moléculaires des maladies liées au vieillissement, Lille, France. [7]Medical Research Council Centre for Neuropsychiatric Genetics and Genomics,, Division of Psychological Medicine and Clinical Neuroscience, School of Medicine, Cardiff University, Cardiff, UK. [8]Department of Neurology, Erasmus Medical Centre, Rotterdam, the Netherlands. [9]Department of Internal Medicine, Erasmus Medical Centre, Rotterdam, the Netherlands. [10]Department of Epidemiology, Erasmus Medical Centre, Rotterdam, the Netherlands. [11]Leiden Academic Centre for Drug Research, Leiden, the Netherlands. [12]Nuffield Department of Population Health Oxford University, Oxford, UK. [13]Medical Research Council Prion Unit at University College London, University College London Institute of Prion Diseases, London, UK. [14]Department of Neurology, II B Sant Pau, Hospital de la Santa Creu i Sant Pau, Universitat Autònoma de Barcelona, Barcelona, Spain. [15]Biomedical Research Networking Center on Neurodegenerative Diseases, National Institute of Health Carlos III, Madrid, Spain. [16]Division of Neurogenetics and Molecular Psychiatry, Department of Psychiatry and Psychotherapy, Faculty of Medicine and University Hospital Cologne, University of Cologne, Cologne, Germany. [17]The John P. Hussman Institute for Human Genomics, University of Miami, Miami, FL, USA. [18]Université Montpellier, INSERM, Institute for Neurosciences of Montpellier, Montpellier, France. [19]Cardiovascular Health Research Unit, Department of Medicine, University of Washington, Seattle, WA, USA. [20]Université Paris-Saclay, Commissariat à l'Énergie Atomique et aux Énergies Alternatives, Centre National de Recherche en Génomique Humaine Evry, Gif-sur-Yvette, France. [21]Experimental Neuro-psychobiology Laboratory, Department of Clinical and Behavioral Neurology, Istituto di Ricovero e Cura a Carattere Scientifico Santa Lucia Foundation, Rome, Italy. [22]Department of Neurodegenerative Science, Van Andel Institute, Grand Rapids, MI, USA. [23]Division of Psychiatry and Behavioral Medicine, Michigan State University College of Human Medicine, Grand Rapids, MI, USA. [24]HudsonAlpha Institute for Biotechnology, Huntsville, AL, USA. [25]Department of Neuroscience, Catholic University of Sacred Heart, Fondazione Policlinico Universitario A. Gemelli Istituto di Ricovero e Cura a Carattere Scientifico, Rome, Italy. [26]Université Bordeaux, INSERM, Bordeaux Population Health Research Center, Bordeaux, France. [27]Department of Neurology, Bordeaux University Hospital, Bordeaux, France. [28]UKDRI Cardiff, School of Medicine, Cardiff University, Cardiff, UK. [29]Department of Biostatistics, Boston University School of Public Health, Boston, MA, USA. [30]Framingham Heart Study, Framingham, MA, USA. [31]Department of Neurology, Boston University School of Medicine, Boston, MA, USA. [32]Department of Epidemiology, Boston University, Boston, MA, USA. [33]Department of Medicine (Biomedical Genetics), Boston University, Boston, MA, USA. [34]Neurogenomics and Informatics Center, Washington University School of Medicine, St Louis, MO, USA. [35]Psychiatry Department, Washington University School of Medicine, St Louis, MO, USA. [36]Hope Center for Neurological Disorders, Washington University School of Medicine, St Louis, MO, USA. [37]Dementia Research Centre, University College London Queen Square Institute of Neurology, London, UK. [38]Fondazione Istituto di Ricovero e Cura a Carattere Scientifico Ca' Granda, Ospedale Policlinico, Milan, Italy. [39]University of Milan, Milan, Italy. [40]Université Brest, INSERM, Etablissement Français du Sang, Centre Hospitalier Universitaire Brest, Unité Mixte de Recherche 1078, GGB, Brest, France. [41]Genome Diagnostics, Department of Human Genetics, VU University, AmsterdamUMC (location VUmc), Amsterdam, the Netherlands. [42]Department of Neurology and Neurological Sciences, Stanford University, Stanford, CA, USA. [43]Department of Epidemiology and Biostatistics, Case Western Reserve University, Cleveland, OH, USA. [44]Clinical and Experimental Science, Faculty of Medicine, University of Southampton, Southampton, UK. [45]Department of Complex Trait Genetics, Center for Neurogenomics and Cognitive Research, Amsterdam Neuroscience, Vrije University, Amsterdam, the Netherlands. [46]McGill University and Genome Quebec Innovation Centre, Montreal, Quebec, Canada. [47]Department of Human Genetics, Amsterdam UMC, University of Amsterdam, Amsterdam Reproduction and Development Research Institute, Amsterdam, the Netherlands. [48]Dr. John T. Macdonald Foundation Department of Human Genetics, University of Miami, Miami, FL, USA. [49]Institute of

Neurology, Catholic University of the Sacred Heart, Rome, Italy. [50]Taub Institute on Alzheimer's Disease and the Aging Brain, Department of Neurology, Columbia University, New York, NY, USA. [51]Gertrude H. Sergievsky Center, Columbia University, New York, NY, USA. [52]Institute of Gerontology and Geriatrics, Department of Medicine and Surgery, University of Perugia, Perugia, Italy. [53]Human Genetics, School of Life Sciences, University of Nottingham, Nottingham, UK. [54]Department of Neuroscience, Psychology, Drug Research and Child Health University of Florence, Florence, Italy. [55]IRCCS Fondazione Don Carlo Gnocchi, Florence, Italy. [56]Penn Neurodegeneration Genomics Center, Department of Biostatistics, Epidemiology, and Informatics, University of Pennsylvania Perelman School of Medicine, Philadelphia, PA, USA. [57]Penn Neurodegeneration Genomics Center, Department of Pathology and Laboratory Medicine, University of Pennsylvania Perelman School of Medicine, Philadelphia, PA, USA. [58]Genomic and Molecular Epidemiology Laboratory, School of Biosciences and Veterinary Medicine, University of Camerino, Camerino, Italy. [59]Université Lille, INSERM, Centre Hospitalier Universitaire Lille, UMR1172, Resources and Research Memory Center (MRRC) of Distalz, Licend, Lille, France. [60]Fundació Docència i Recerca MútuaTerrassa and Movement Disorders Unit, Department of Neurology, University Hospital MútuaTerrassa, Barcelona, Spain. [61]Memory Disorders Unit, Department of Neurology, Hospital Universitari Mutua de Terrassa, Barcelona, Spain. [62]Université de Nantes, Centre Hospitalier Universitaire Nantes, Centre National de la Recherche Scientifique, INSERM, l'institut du Thorax, Nantes, France. [63]Institute of Social Medicine, Occupational Health and Public Health, University of Leipzig, Leipzig, Germany. [64]Neurology Service, Marqués de Valdecilla University Hospital (University of Cantabria and IDIVAL), Santander, Spain. [65]Laboratory for Advanced Hematological Diagnostics, Department of Hematology and Stem Cell Transplant, Lecce, Italy. [66]Department of Psychiatry and Glenn Biggs Institute for Alzheimer's and Neurodegenerative Diseases, San Antonio, TX, USA. [67]Laboratory of Neuropsychiatry, Department of Clinical and Behavioral Neurology, Istituto di Ricovero e Cura a Carattere Scientifico Santa Lucia Foundation, Rome, Italy. [68]Department of Neurodegenerative Diseases and Geriatric Psychiatry, University Hospital Bonn, Medical Faculty, Bonn, Germany. [69]German Center for Neurodegenerative Diseases, Bonn, Germany. [70]Université Rouen Normandie, INSERM U1245 and CHU Rouen, Department of Neurology and CNRMAJ, Rouen, France. [71]Memory and Aging Center, Department of Neurology, University of California, San Francisco, CA, USA. [72]Reta Lila Weston Research Laboratories, Department of Molecular Neuroscience, University College London Institute of Neurology, London, UK. [73]Cluster of Excellence Cellular Stress Responses in Aging-Associated Diseases, University of Cologne, Cologne, Germany. [74]These authors contributed equally: Henne Holstege, Marc Hulsman, Camille Charbonnier, Gaël Nicolas, Céline Bellenguez, Jean-Charles Lambert. ✉e-mail: h.holstege@amsterdamumc.nl; m.hulsman1@amsterdamumc.nl; gaelnicolas@hotmail.com; jean-charles.lambert@pasteur-lille.fr

## Methods

In-depth descriptions of all methods are described in Methods section of the Supplementary Note.

### Sample processing, genotype calling and QC

We collected the exome, whole genome sequencing (WGS) or exome extract sequencing data of a total of 52,361 individuals, brought together by the Alzheimer Disease European Sequencing (ADES) consortium, the Alzheimer's Disease Sequencing Project (ADSP)[21] and several independent study cohorts (Supplementary Table 1). Exome extract samples only contained the raw reads that cover the ten genes identified in stage 1. Across all cohorts, AD cases were defined according to National Institute on Aging-Alzheimer's Association criteria[22] for possible or probable AD or according to National Institute of Neurological and Communicative Disorders and Stroke-Alzheimer's Disease and Related Disorders Association criteria[23] depending on the date of diagnosis. When possible, supportive evidence for an AD pathophysiological process was sought (including cerebrospinal fluid biomarkers) or the diagnosis was confirmed by neuropathological examination (Supplementary Table 1). AD cases were annotated with the age at onset or age at diagnosis (2,014 samples); otherwise, samples were classified as late-onset AD (366 samples). Controls were not diagnosed with AD. All contributing datasets were sequenced using a paired-end Illumina platform; different exome capture kits were used and a subset of the sample was sequenced using WGS (Supplementary Table 2).

A uniform pipeline was used to process both the stage 1 and stage 2 datasets. Raw sequencing data from all studies were processed relative to the GRCh37 reference genome, the read alignments of possible chimeric origin were filtered and a GATK-based pipeline was used to call variants, while correcting for estimated sample contamination percentages. Samples were included in the datasets after they passed a stringent QC pipeline: samples were removed when they had high missingness, high contamination, a discordant genetic sex annotation, non-European ancestry, high numbers of new variants (with reference to dbSNP v.150), deviating heterozygous/homozygous or transition/transversion ratios. Further, we removed family members up to the third degree and individuals who carried a pathogenic variant in *PSEN1*, *PSEN2*, *APP* or in other genes causative for Mendelian dementia diseases (stage 1-only) or when there was clinical information suggestive of non-AD dementia. Variants considered in the analysis also passed a stringent QC pipeline: multiallelic variants were split into biallelic variants; variants that were in complete linkage and near each other were merged. Further, we removed variants that had indications of an oxo-G artifact, were located in short tandem repeat and/or low copy repeat regions, had a discordant balance between reads covering the reference and alternate allele, had a low depth for alternate alleles, deviated significantly from Hardy–Weinberg equilibrium, were considered false positives based on GATK variant quality score recalibration or were estimated to have a batch effect. Variants with >20% genotype missingness (read depth < 6) and differential missingness between the EOAD, LOAD and control groups were removed. To account for uncertainties resulting from variable read coverage between samples, we analyzed variants according to genotype posterior likelihoods, that is, the likelihood of being homozygous for the reference allele and heterozygous or homozygous for the alternate allele. To account for genotype uncertainty, the burden test was performed multiple times with independently sampled genotypes and the average *P* value across these tests is reported.

### Variant prioritization and thresholds

We selected variants in autosomal protein-coding genes that were part of the Ensembl basic set of protein-coding transcripts (Gencode v.19/v.29 (ref. [24]); Supplementary Note) and that were annotated by the Variant Effect Predictor v.94.542 (ref. [25]). Only protein-coding missense and LOF variants were considered (LOF: nonsense, splice acceptor/ donor or frameshifts). Missense and LOF variants were required to have a 'moderate' and 'high' variant effect predictor impact classification, respectively. Then, missense variants were prioritized using REVEL[26], annotation obtained from dbNSFP4.1a[27] and LOF variants were prioritized using LOFTEE v.1.0.2 (ref. [28]). For the analysis, we considered only missense variants with a REVEL score ≥ 25 (score range 0–100) and LOF variants were annotated as 'high confidence' by LOFTEE. Variants were required to have at least 1 carrier (that is, at least 1 sample with a posterior dosage >0.5) and an MAF < 1%, both in the considered dataset and the Genome Aggregation Database v.2.1 populations (non-neurological set).

### Gene burden testing

The burden analysis was based on four deleteriousness thresholds by incrementally including variants from categories with lower levels of predicted variant deleteriousness: LOF; LOF + REVEL ≥ 75; LOF + REVEL ≥ 50; and LOF + REVEL ≥ 25, respectively. This allowed us to identify the variant threshold providing maximum evidence for a differential burden signal. To infer any dependable signal for a specific deleterious threshold, a minimum of 10 damaging alleles appertaining to this deleteriousness threshold was required, that is, a cMAC ≥ 10. Multiple testing correction was performed across all performed tests (up to four per gene). Burden testing was implemented using ordinal logistic regression. This enabled burden testing to particularly weight EOAD cases since previous findings indicated that high-impact variants are enriched in early-onset (EOAD) cases relative to late-onset (LOAD) cases[8]. This implies that the burden of high-impact deleterious genetic variants is ordered according to $burden_{EOAD} > burden_{LOAD} > burden_{control}$. Ordinal logistic regression enabled optimal identification of such signals, while also allowing the detection of EOAD-specific burdens ($burden_{EOAD} > burden_{LOAD} \sim burden_{control}$) and regular case-control signals ($burden_{EOAD} \sim burden_{LOAD} > burden_{control}$). For protective burden signals, the order of the signals is reversed, that is, $burden_{EOAD} < burden_{LOAD} < burden_{control}$. We considered an additive model while correcting for six population covariates, estimated after removal of population outliers. *P* values were estimated using a likelihood-ratio test. Genes were selected for confirmation in stage 2 if the FDR for AD association was <0.1 in stage 1 (Benjamini–Hochberg procedure[29]). For the GWAS-targeted analysis, a more stringent threshold was used (FDR < 0.05) due to the absence of a separate confirmation stage. For the meta-analysis, genes were considered significantly associated with AD when the corrected *P* was <0.05 after family-wise correction using the Holm–Bonferroni procedure[30]. Effect sizes (ORs) of the ordinal logistic regression can be interpreted as weighted averages of the OR being an AD case versus control and the OR being an early-onset AD case or not. To aid interpretation, we additionally estimated 'standard' case/control ORs across all samples per age category (EOAD versus controls and LOAD versus controls) and for age-at-onset categories ≤65 (EOAD), 65–70, 70–80 and >80 using multinomial logistic regression, while correcting for 6 PCA covariates.

### GWAS driver gene identification

For the 75 loci identified in the most recent GWAS[2], genes were selected for burden testing based on earlier published gene prioritizations. First, gene prioritizations were obtained from Schwarzentruber et al.[31] for 33 known loci. For 28 remaining loci, we obtained the tier 1 prioritization from Bellenguez et al.[2]; for loci without prioritization candidates (14 loci), we selected the nearest gene. In total, 81 protein-coding genes were selected (Supplementary Table 7), of which 67 genes had sufficient damaging allele carriers to be tested for at least 1 variant selection threshold. Gene burden testing was performed as described above and multiple testing correction to identify potential driver genes was performed using the Benjamini–Hochberg procedure, with a cutoff of 5%.

## Validation of variant selection

We validated the REVEL variant impact prediction for missense and the LOFTEE impact prediction for LOF variants for all variants with an MAF < 1%, for which there were at least 15 damaging allele carriers. For protein-modifying variants that were not in the most significant burden selection of a gene due to a low predicted impact, we investigated whether they, nevertheless, showed a significant AD association (based on a case/control analysis using logistic regression). Vice versa, for variants that were in the burden selection, we investigated whether their effect size was significantly reduced or oppositely directed from other missense or LOF variants in the burden selection (Fisher's exact test). Individual variant effects were analyzed in the stage 1 dataset, followed by a confirmation analysis in the stage 2 dataset. Multiple testing correction was performed per gene, with an FDR < 0.1 used as the threshold for stage 1 and Holm–Bonferroni ($P < 0.05$) for stage 2.

## Descriptive measures

A variant carrier was defined as an individual for whom the summed dosage of all the variants in the considered variant deleteriousness category is ≥0.5 (see Methods section in the Supplementary Note). Carrier frequencies (CFs) were determined as the number of carriers/number of total samples. Attributable fraction for cases in an age group was estimated as the probability of a case with an age at onset in the age window $i$ being exposed to a specific gene burden ($CF_{case,gene,i}$), multiplied by an estimate of the attributable fraction among the exposed for these cases: $\left(\frac{OR_{gene,i}-1}{OR_{gene,i}}\right)$ (with the OR being an approximation of the relative risk)[32,33]. For large effect sizes, this estimate approaches the difference in carrier frequency between cases and controls: $(CF_{case,gene,i}) - (CF_{control,gene})$.

## Sensitivity analyses

We determined if the observed effects could be explained by age differences between cases and controls. We constructed an age-matched sample, dividing samples into strata based on age/age at onset, with each stratum covering 2.5 years. Case/control ratios in all strata were kept between 0.1 and 10 by downsampling controls or cases, respectively. Subsequently, samples were weighted using the 'propensity weighting within strata method' (Supplementary Note). Finally, a case-control logistic regression was performed both on the unweighted and weighted case-control labels and estimated ORs and confidence intervals (CIs) were compared (Extended Data Fig. 8) Also, we determined if somatic mutations due to age-related clonal hematopoiesis could have confounded the results. We calculated for all heterozygous calls in the burden selection the balance between reference and alternate reads and compared these to reference values (Supplementary Table 14). While APOE was not included as a confounder, we performed a separate APOE interaction analysis (Supplementary Table 13) through a likelihood-ratio test between a model label ~ gene_burden_score + APOE_e4_dosage and an interaction model label ~ gene_burden_score + APOE_e4_dosage + APOE_e4_dosage ×gene_burden_score. This test was performed on a reduced dataset, from which datasets in which APOE status was used as the selection criterion were removed.

## Power analysis

Power calculations were performed for ordinal and Firth logistic regression (case-control and EOAD versus rest; Fig. 1b and Supplementary Table 6). Given the ORs for the EOAD and LOAD cases, and the cMAC per gene, we sampled the number of alleles in the EOAD cases, LOAD cases and controls according to a multinomial distribution. We randomized these allele carriers across the dataset and performed the burden test as described above. The power for genes with a cMAC < 10 was set to 0 since these genes were not analyzed.

## Reporting summary

Further information on research design is available in the Nature Research Reporting Summary linked to this article.

## Data availability

The genetic variants analyzed in this study are listed in the Supplementary Data attached to this article. Summary statistics of the stage 1 analysis are publicly available at Zenodo (https://doi.org/10.5281/zenodo.6818051)[34] and they can also be downloaded from https://holstegelab.eu/data/. For all tests with a cMAC ≥10, this includes Ensembl gene ID, gene name, variant category, cMAC, P value, beta and s.e.m. The ADSP dataset, which includes the ADNI dataset used in this analysis, is publicly available on request from https://dss.niagads.org/datasets/. The accession numbers of the data used in this analysis are: ADSP DBGap: phs000572.v7.p4 (stage 1); ADSP NIAGADS: https://dss.niagads.org/datasets/ng00067-v2/ (stage 2). Source data to Figs. 2 and 3 are published alongside this paper.

## Code availability

The software and algorithms used in the analysis are described in the Supplementary Note attached to this Letter. Self-contained code v.0.1.0 can be accessed at https://github.com/holstegelab/shortread_seq_analysis and Zenodo (https://doi.org/10.5281/zenodo.6827458)[35].

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

## Acknowledgements

We thank all the study participants, their families, the participating medical staff, general practitioners, pharmacists and all laboratory personnel involved in patient diagnosis, blood collection, blood biobanking, DNA preparation and sequencing. The work in this manuscript was carried out on the Cartesius supercomputer, which is embedded in the Dutch national e-infrastructure with the support of the SURF Cooperative. Computing hours were granted in 2016, 2017, 2018 and 2019 to H. Holstege by the Dutch Research Council (project name: 100-plus; project nos. 15318 and 17232). This research was conducted using the funding obtained by the following study cohorts: ADES-FR, AgeCoDe-UKBonn; Barcelona SPIN; AC-EMC; ERF and Rotterdam; ADC-Amsterdam; 100-plus study; EMIF-90+; Control Brain Consortium; PERADES; StEP-AD; Knight-ADRC; UCSF/NYGC/UAB; UCL-DRC EOAD; ADSP. Data used in preparation of this article were obtained from the Alzheimer's Disease Neuroimaging Initiative (ADNI) database (https://adni.loni.usc.edu/). The investigators within ADNI are listed as supplementary authors and can be found in Section 5 of the Supplementary Note. Full consortium acknowledgements and funding sources are listed in Section 4 of the Supplementary Note.

## Author contributions

H.Holstege, G.N. and J.-C.L. jointly supervised the research. H.Holstege, M.H., C.Charbonnier, B.G.-B., O.Q., G.N., C.Bellenguez and J.-C.L. were the core writing and analysis group. H.Holstege, M.H., C.Charbonnier, B.G.-B., O.Q., D.Grozeva, J.G.J.v.R., R.S., S.A., N.A., P.J.N., O.D.-I., H.Hummerich2, A.K., J.C., J.C.v.S., J.H., A.R., S.M. W.M.v.d.F., C.M.v.D, J.W., G.N., C.Bellenguez and J.-C.L. were the ADES cohort working group. M.H., S.J.v.d.L., M.J.T.R., N.T. and H.Holstege represented the 100-plus study and Netherlands Brain Bank cohorts and contributed to sample collection. P.J.V. represented the EMIF-AD-90 study cohort and contributed to sample collection. J.G.J.v.R., M.O.M., J.C.v.S. represented the AC-EMC cohort and contributed to sample collection. M.H., S.J.v.d.L., F.B., B.T., A.W.L., I.E.J., W.M.v.d.F., P.S. and H.Holstege represented the ADC-Amsterdam cohort and contributed to sample collection. A.K., S.G.R.-H., M.W. and A.R. represented the AgeCoDe-UKBonn cohort and contributed to sample collection. C.Charbonnier, O.Q., D.W., A.Z., D.C., A.-C.R., S.R., G.N., A.B., J.-F.Deleuze, M.L., F.P., E.G., J.-F.Dartigues, R.R., S.D., B.G.-B., C.Berr, C.Bellenguez, J.-C.L. and P.A. represented the ADES-France cohort and contributed to sample collection. A.C.N., L.A.F., J.L.H., R.Mayeux, M.A.P.-V., J.C.B., L.-S.W., G.W.B., A.L.D.S., E.R.M., S.Seshadri and G.D.S. represented the ADSP cohort and contributed to sample collection. M.H., S.J.v.d.L., E.A.S., D.Sie, J.J.P.G., M.M.A.M.M., R.v.S. and H.Holstege represented the Amsterdam-UMC cohort and contributed to sample collection. O.D.-I., A.L. and J.C. represented the Barcelona SPIN cohort and contributed to sample collection. N.C.F., J.B., R.G. and J.H. represented the Control Brain Consortium cohort and contributed to sample collection. M.V.F. and C.Cruchaga represented the Knight-ADRC cohort and contributed to sample collection. D.Grozeva, R.R., S.Saad, N.D., A.M., R.Marshall, L.L., A.D., B.N., P.B., C.M., C.H., D.Galimberti, D.Seripa, P.M., S.Sorbi, G.S., K.M., P.S.-J., P.P., R.S. and J.W. represented the PERADES cohort and contributed to sample collection. S.A., N.A., R.K., A.G.U., M.A.I., F.R., M.K.I. and C.M.v.D. represented the Rotterdam and ERF cohorts and contributed to sample collection. Y.L.G., V.N., K.M. and M.D.G. represented the StEP-AD cohort and contributed to sample collection. H.Hummerich, P.J.N., N.S.R., J.M.S. and S.M. represented the UCL-DRC EOAD cohort. J.N.C., R.M.M. and J.S.Y. represented the UCSF/NYGC/UAB cohort and contributed to sample collection.

## Competing interests

The authors declare no competing interests.

## Additional information

**Extended data** is available for this paper at https://doi.org/10.1038/s41588-022-01208-7.

**Correspondence and requests for materials** should be addressed to Henne Holstege, Marc Hulsman, Gaël Nicolas or Jean-Charles Lambert.

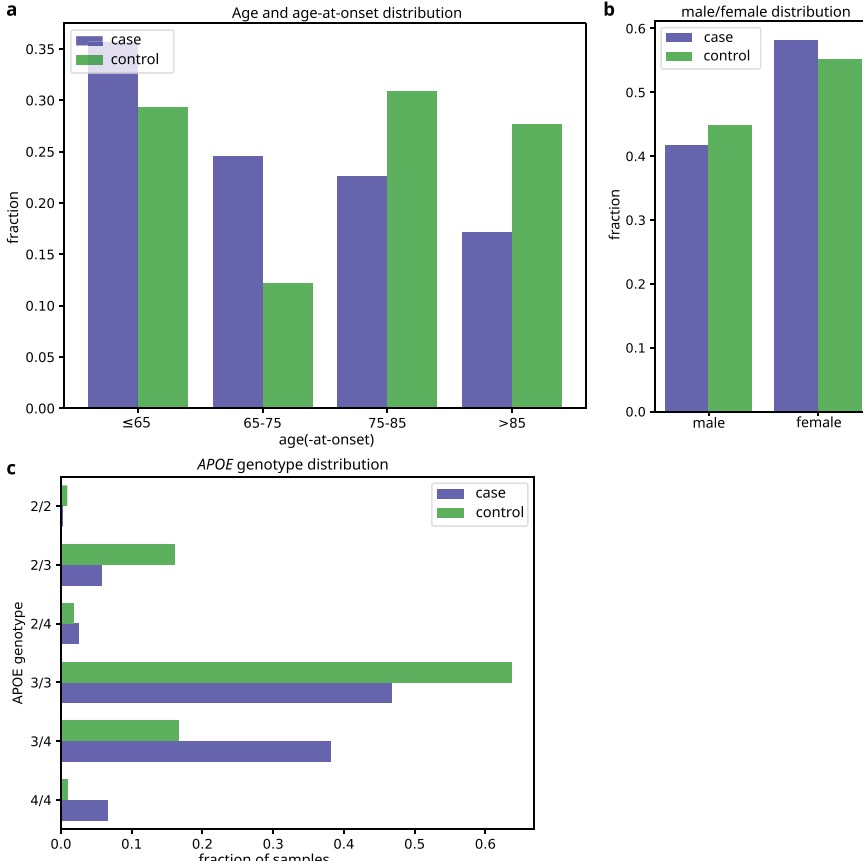

**Extended Data Fig. 1 | Age, gender, APOE genotype distribution.** Age, gender and APOE genotype distribution of all samples, stratified by case/control status.

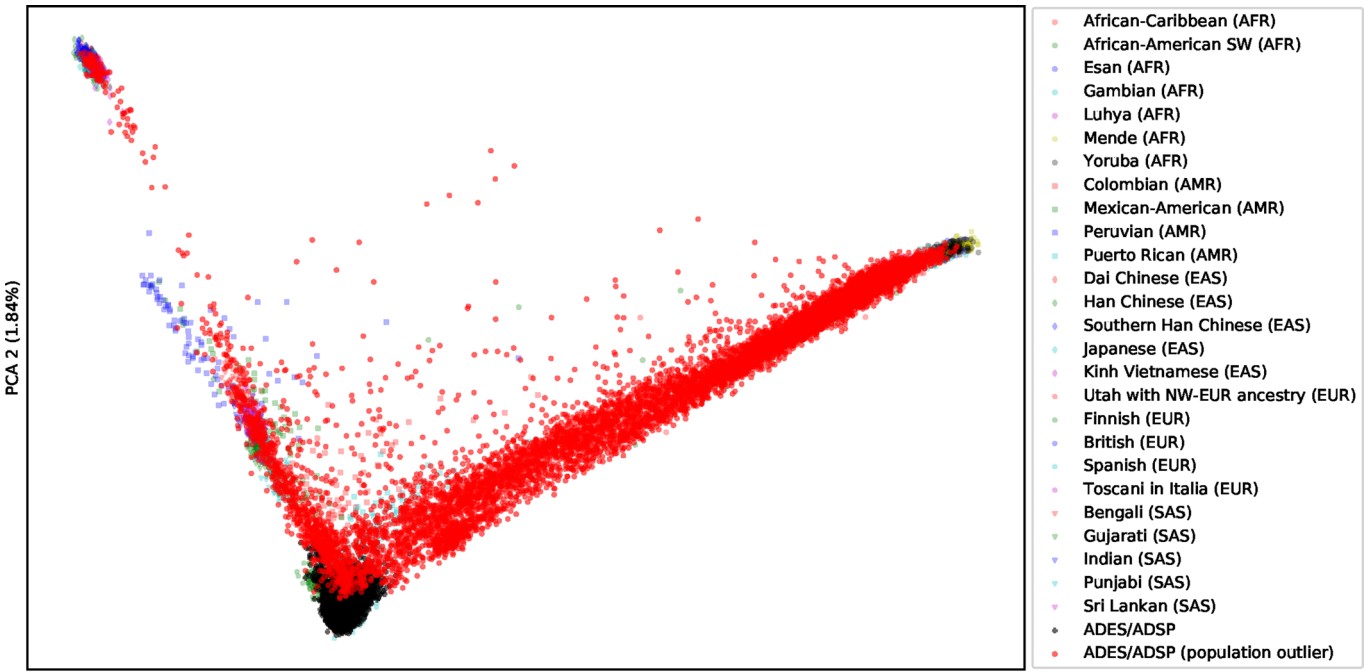

**Extended Data Fig. 2 | PCA: Sample population compared to 1,000 G population samples. Sample population compared to 1,000 G population samples**. First two PCA components of the study samples used for the Stage 1 and Stage 2 analysis, shown in context of the 1000 Genomes samples for reference (see Supplementary Note section 1.3.4). Samples in red are considered population outliers. Samples with only exome-extracts were not included in this analysis.

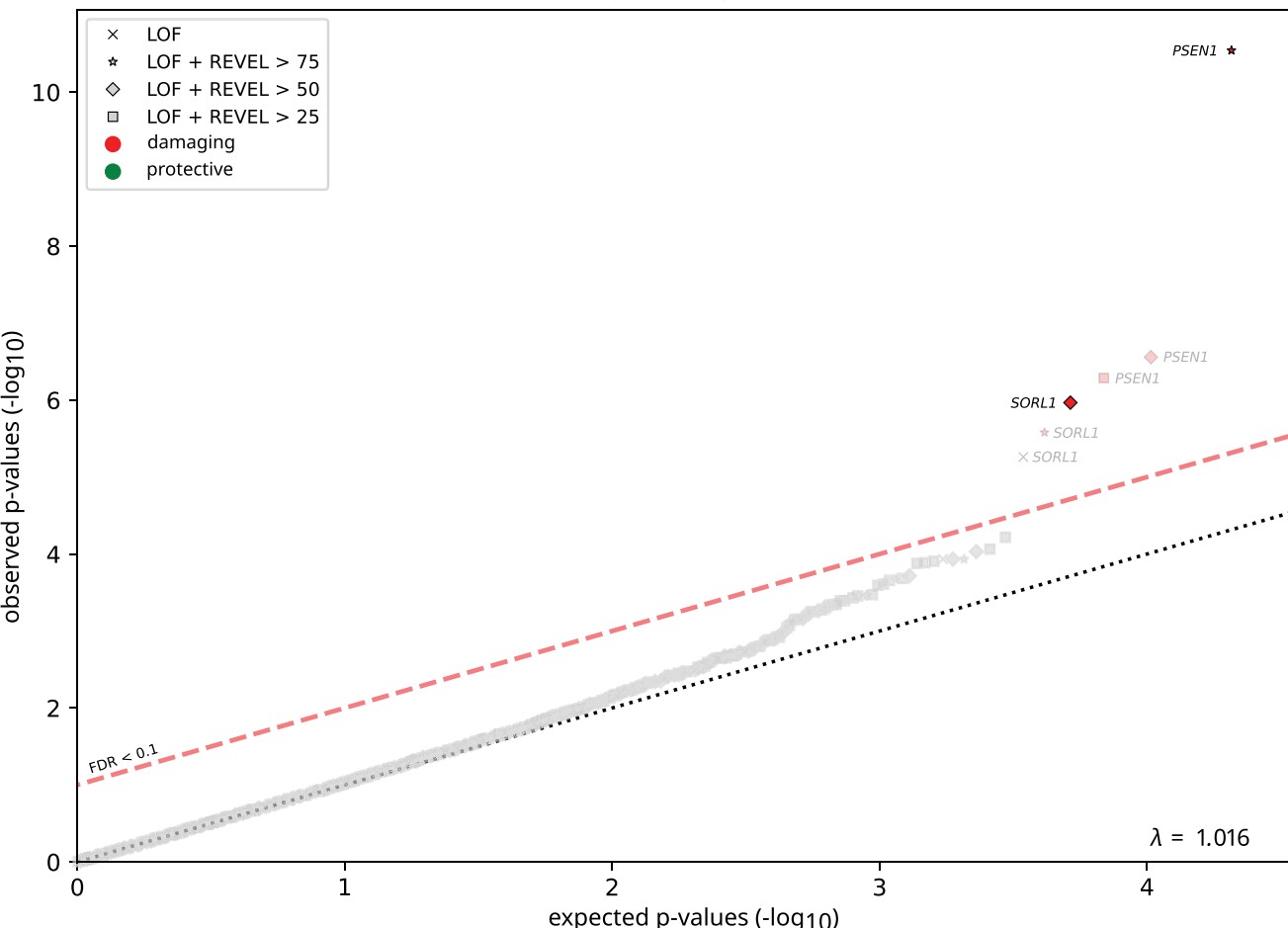

Quantile-quantile plot for stage 2

**Extended Data Fig. 3 | P value inflation in Stage-2 analysis. P value inflation in Stage-2 analysis:** Quantile-quantile plot for Stage-2 (without exome-extract samples), based on a ordinal logistic burden test (see Methods). Results are shown for all burden tests (n = 20,681) for which at least 10 damaging alleles were present in this dataset (based on 4 different variant deleteriousness thresholds per gene). While not used in this analysis, the threshold for multiple testing correction based on FDR < 0.1 is shown for reference. The genomic p-value inflation was 1.016. Note that causative mutations were not separately removed in Stage-2, as we focused on a specific set of genes.

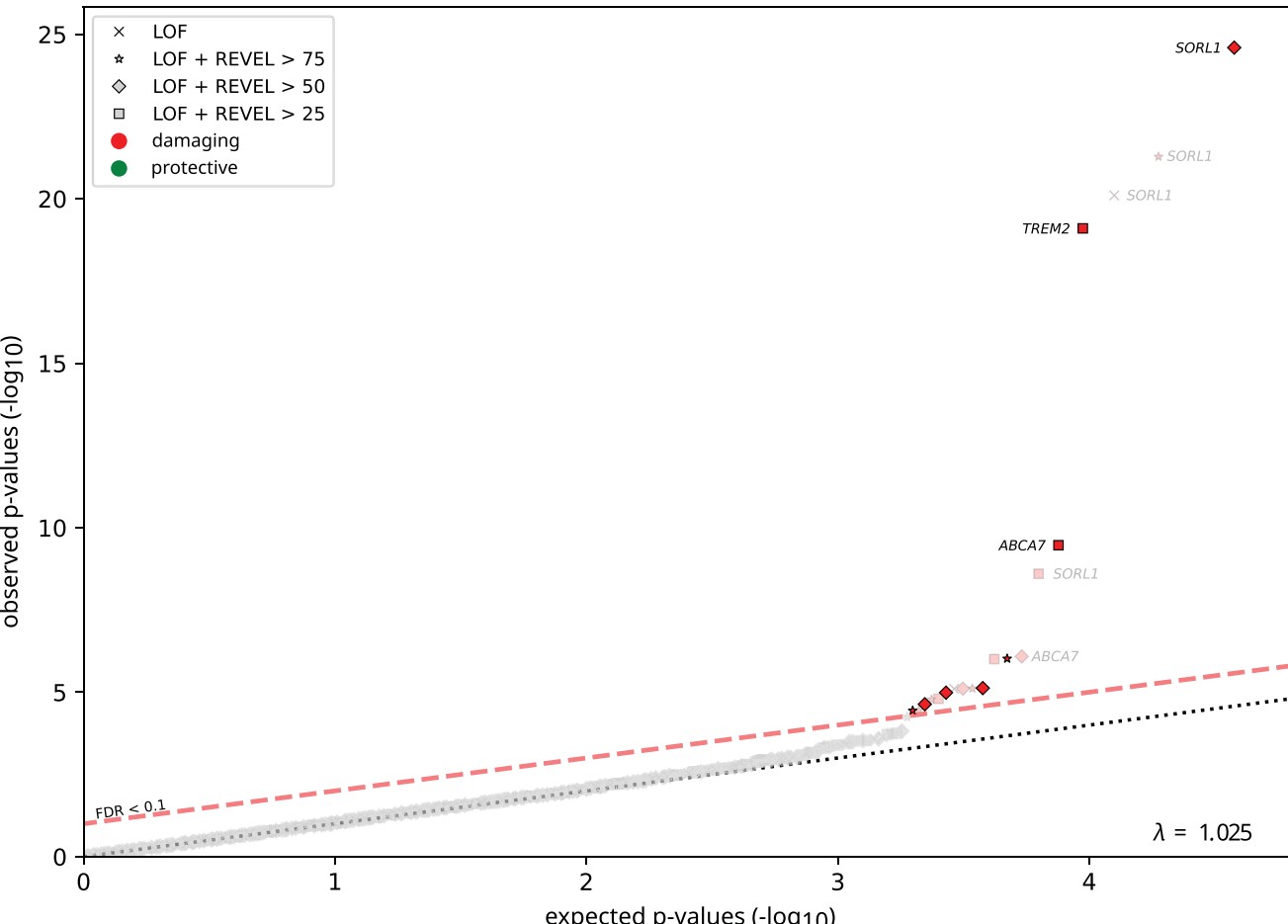

**Extended Data Fig. 4 | P value inflation in the mega-analysis dataset.**
**P value inflation in the mega-analysis dataset:** Quantile-quantile plot for the mega-analysis dataset (without exome-extract samples) based on a ordinal logistic burden test (see Methods). Results are shown for all burden tests (n = 37,710) for which at least 10 damaging alleles were present in this dataset (based on 4 different variant deleteriousness thresholds per gene). For reference, the threshold for multiple testing correction based on a false discovery rate threshold of 0.1 is shown. P values for the mega-analysis are shown in Supplementary Table 15. The genomic p-value inflation was 1.025.

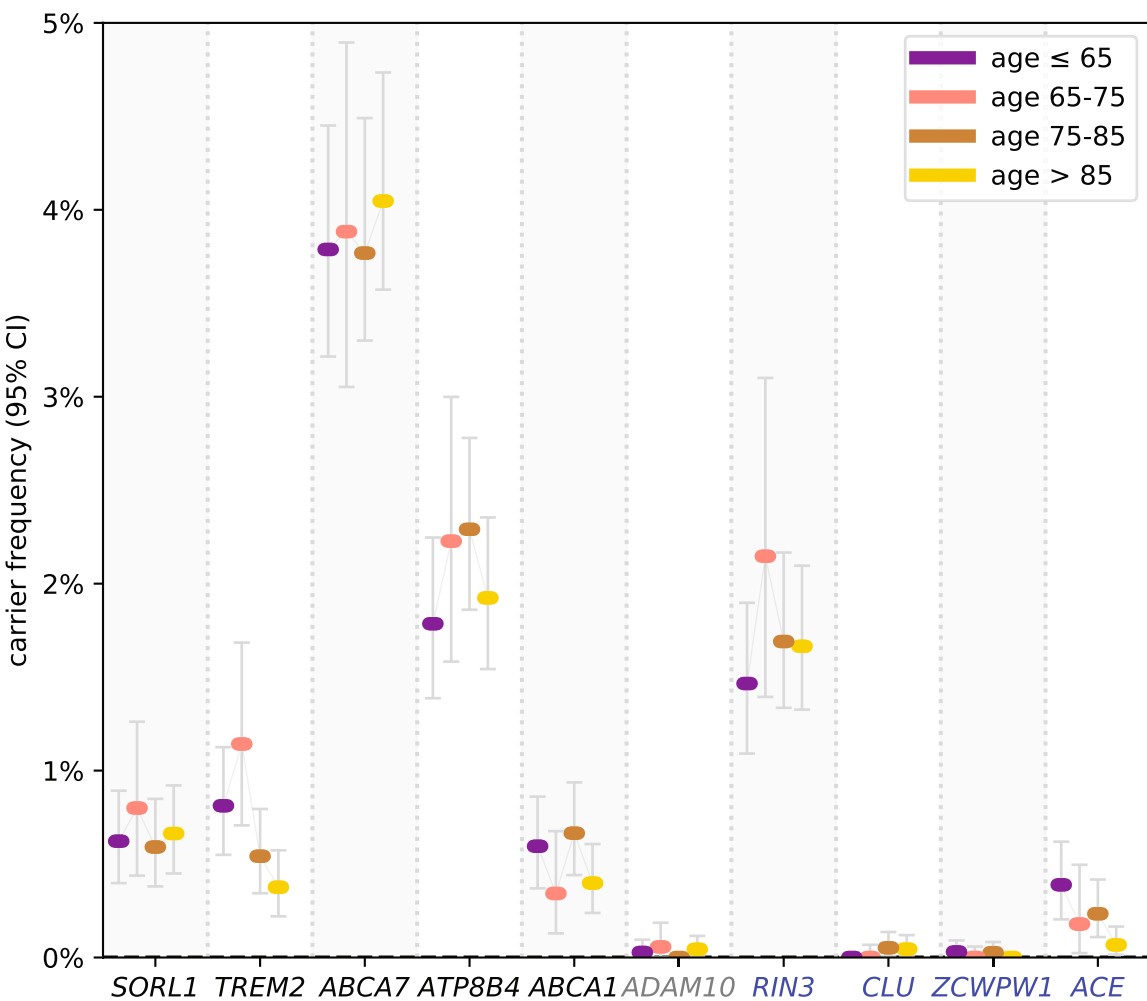

**Extended Data Fig. 5 | Variant carrier frequency in controls by age last seen. Variant carrier frequency in controls by age last seen**: Carrier frequency in controls by age last seen for the variant selection threshold with the strongest association, as observed in the mega-analysis (n = 31,905 unique individuals); *RIN3, CLU, ZCWPW1, ACE* (n = 29,727 unique individuals; that is without exome-extracts) (Table 3, refined). Black: genes significant in the meta-analysis. Grey: genes not significant in meta-analysis. Blue: genes detected in the GWAS targeted analysis.

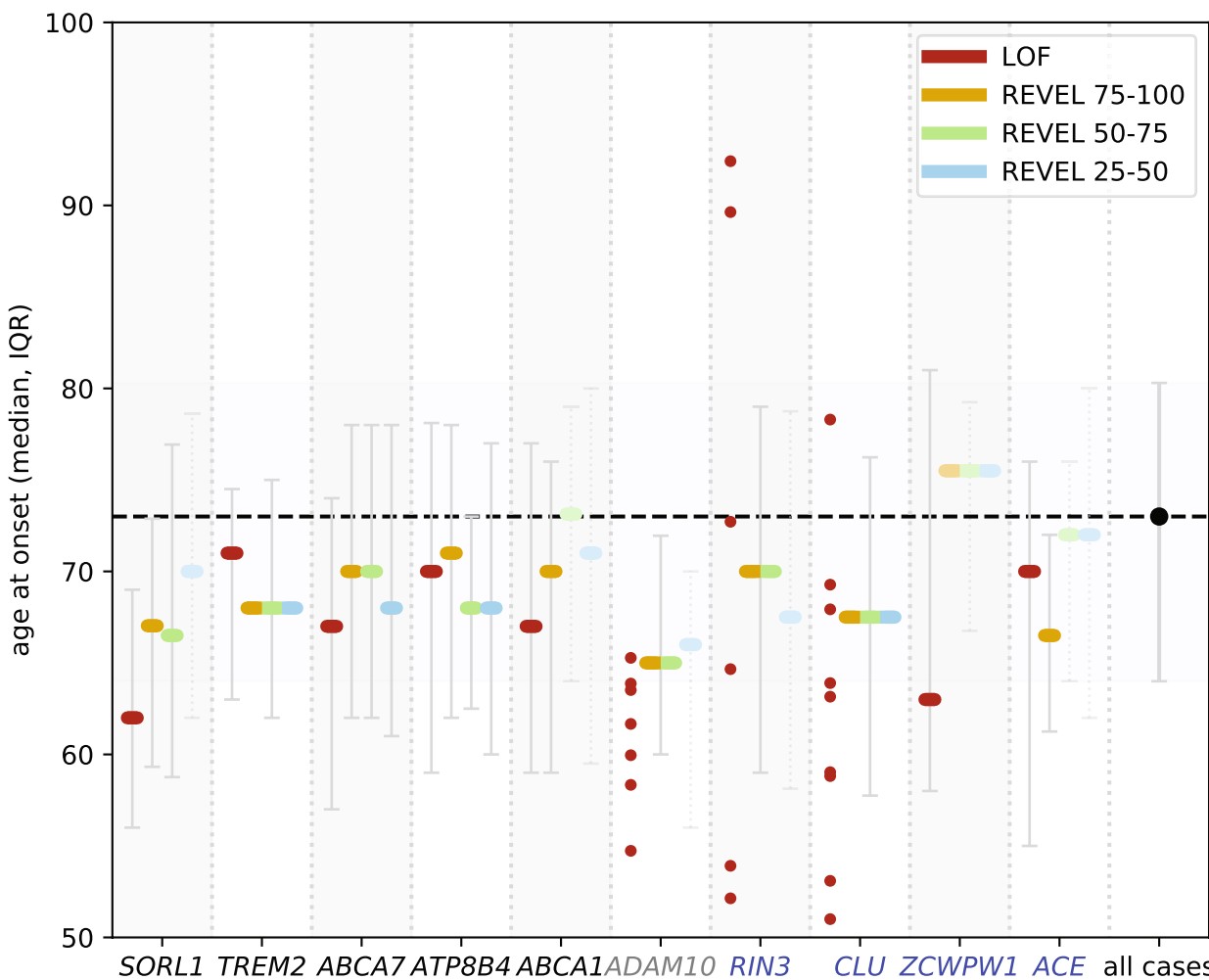

**Extended Data Fig. 6 | Age-at-onset by variant deleteriousness category. Age-at-onset by variant deleteriousness category:** Age-at-onset (median and IQR) in the mega-analysis (n = 31,905 unique individuals); *RIN3, CLU, ZCWPW1, ACE* (n = 29,727 unique individuals; that is without exome-extracts). Samples in variant deleteriousness categories with <10 samples are shown individually. The median age at onset and IQR for the complete mega-analysis dataset is shown on the right. Black: genes significant in the meta-analysis. Grey: genes not significant in meta-analysis. Blue: genes detected in the GWAS targeted analysis.

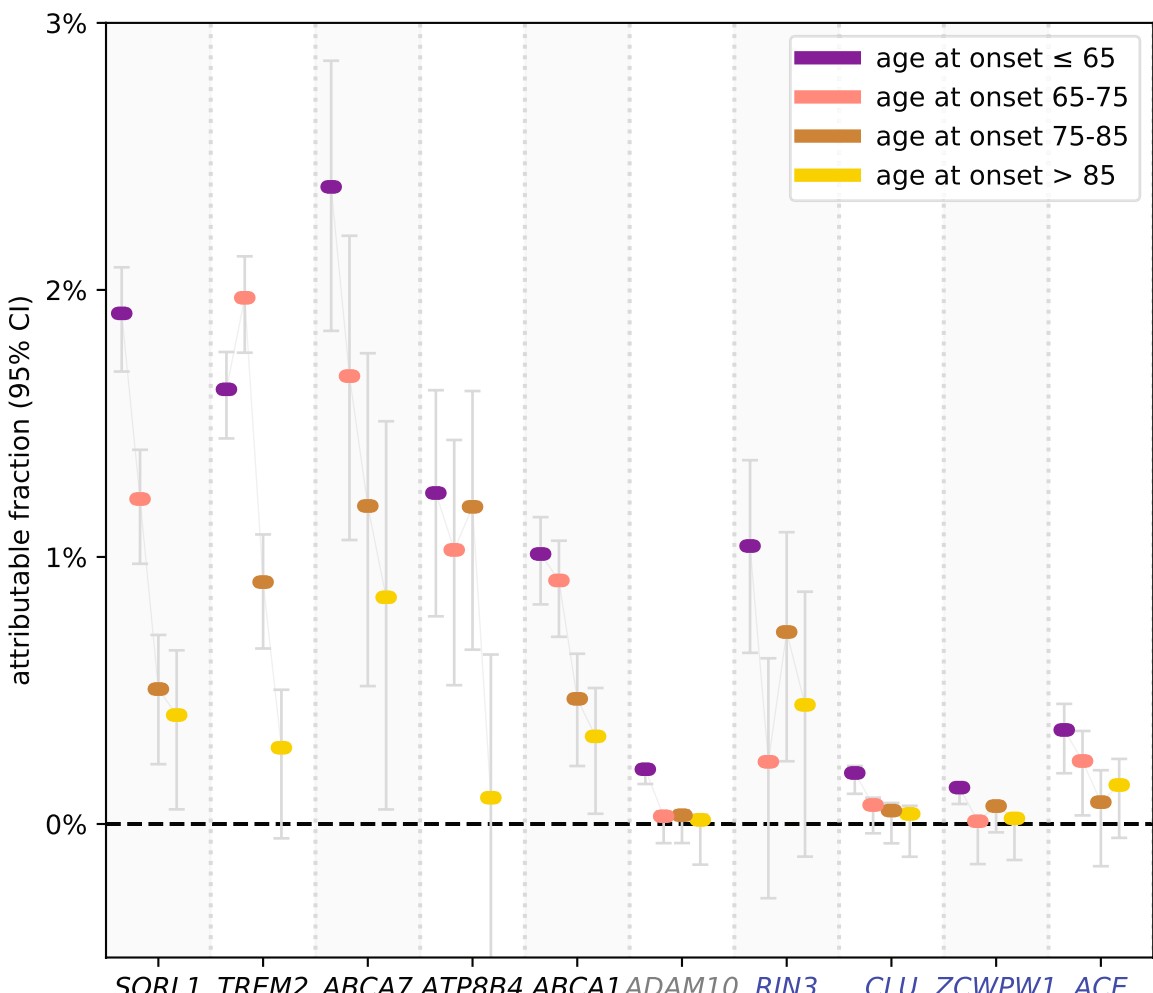

**Extended Data Fig. 7 | Attributable fraction per gene and age-at-onset category. Attributable fraction per gene and age-at-onset category:** Attributable fractions as derived based on the mega-analysis in the mega-analysis (n = 31,905 unique individuals); *RIN3, CLU, ZCWPW1, ACE* (n = 29,727 unique individuals; that is without exome-extracts). The attributable fraction of a gene is an estimate of the fraction of AD cases in a specific age group that have become part of this dataset due to carrying a rare damaging variant in the respective gene (Methods). This estimate accounts only for variants in the burden selection. Black: genes significant in the meta-analysis. Grey: genes not significant in meta-analysis. Blue: genes detected in the GWAS targeted analysis.

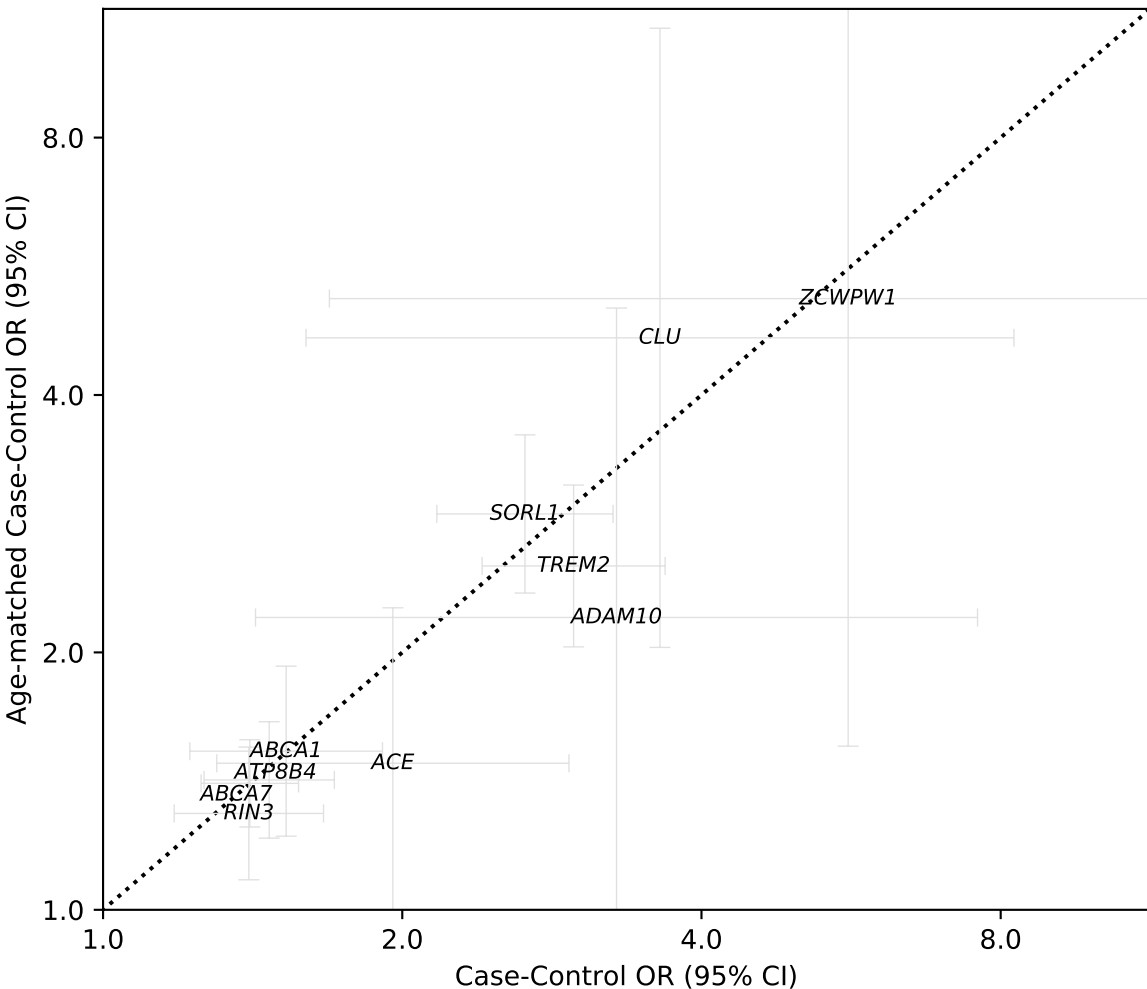

**Extended Data Fig. 8 | Sensitivity Analysis: AD vs Age association. AD vs Age association:** Sensitivity analysis of the gene burden tests (for the most significant deleteriousness thresholds, Table 2) for the mega-analysis dataset (*RIN3, CLU, ZCWPW1, ACE*: without exome-extracts) (respectively n = 31,905 and n = 29,727 unique individuals). Comparison of the case/control odds ratio of an age-matched and a non-age-matched analysis. Age-matching was performed as described in the methods. Based on the confidence intervals, we cannot exclude that the signals in *ACE*, *ADAM10* and *ZCWPW1* are affected by other age-related conditions. Note however, that the signals in *ADAM10* and *ZCWPW1* are based on very few variants, such that confidence intervals are expected to be wide.

Henne Holstege
Marc Hulsman
Gaël Nicolas

# Reporting Summary

## Statistics

For all statistical analyses, confirm that the following items are present in the figure legend, table legend, main text, or Methods section.

| n/a | Confirmed | |
|---|---|---|
| ☐ | ☒ | The exact sample size (*n*) for each experimental group/condition, given as a discrete number and unit of measurement |
| ☐ | ☒ | A statement on whether measurements were taken from distinct samples or whether the same sample was measured repeatedly |
| ☐ | ☒ | The statistical test(s) used AND whether they are one- or two-sided *Only common tests should be described solely by name; describe more complex techniques in the Methods section.* |
| ☐ | ☒ | A description of all covariates tested |
| ☐ | ☒ | A description of any assumptions or corrections, such as tests of normality and adjustment for multiple comparisons |
| ☐ | ☒ | A full description of the statistical parameters including central tendency (e.g. means) or other basic estimates (e.g. regression coefficient) AND variation (e.g. standard deviation) or associated estimates of uncertainty (e.g. confidence intervals) |
| ☐ | ☒ | For null hypothesis testing, the test statistic (e.g. *F*, *t*, *r*) with confidence intervals, effect sizes, degrees of freedom and *P* value noted *Give P values as exact values whenever suitable.* |
| ☒ | ☐ | For Bayesian analysis, information on the choice of priors and Markov chain Monte Carlo settings |
| ☒ | ☐ | For hierarchical and complex designs, identification of the appropriate level for tests and full reporting of outcomes |
| ☐ | ☒ | Estimates of effect sizes (e.g. Cohen's *d*, Pearson's *r*), indicating how they were calculated |

*Our web collection on statistics for biologists contains articles on many of the points above.*

## Software and code

Policy information about availability of computer code

| Data collection | No software was used. |
|---|---|
| Data analysis | BCFtools (version 1.8)<br>Bedtools (version 2.26.0)<br>BWA (version 0.7.15-r1140)<br>EAGLE (version 2.4)<br>GATK (version 3.8-1): BQSR, HaplotypeCaller, VQSR, CombineGVCFs, GenotypeGVCFs<br>LOFTEE (version 1.0.2)<br>MACS (version 1.4)<br>Matplotlib (version 3.1.1)<br>Numpy (version 1.17.4)<br>Picard tools (version 2.10.5)<br>Python (version 2.7, 3.6)<br>R (version 3.4.3)<br>R-MASS package (version 7.3-51.5)<br>R-lmtest package (version 0.9-35)<br>VEP (version 94.5)<br>VerifyBamID2 (version 1.0.5)<br>Samblaster (version 0.1.24)<br>Samtools (version 1.8)<br>Scipy (version 1.4.1)<br>Seekin (version 1.0)<br>SKLearn (version 0.20.3) |

Custom scripts: https://github.com/holstegelab/shortread_seq_analysis
https://doi.org/10.5281/zenodo.6827458

For manuscripts utilizing custom algorithms or software that are central to the research but not yet described in published literature, software must be made available to editors and reviewers. We strongly encourage code deposition in a community repository (e.g. GitHub). See the Nature Portfolio guidelines for submitting code & software for further information.

## Data

Policy information about availability of data

All manuscripts must include a data availability statement. This statement should provide the following information, where applicable:
- Accession codes, unique identifiers, or web links for publicly available datasets
- A description of any restrictions on data availability
- For clinical datasets or third party data, please ensure that the statement adheres to our policy

The genetic variants analyzed during this study are listed in the Supplementary Data attached to this Letter.
Summary statistics of the discovery analysis were deposited to the Zenodo Digital Archive: 10.5281/zenodo.6818051
The ADSP dataset (which includes the ADNI dataset) used in this analysis is publicly available upon request: https://dss.niagads.org/datasets/
Accession numbers of data used in this analysis:
ADSP DBGap: phs000572.v7.p4 (stage-1)
ADSP Niagads: ng00067.v2 (stage-2)

# Field-specific reporting

Please select the one below that is the best fit for your research. If you are not sure, read the appropriate sections before making your selection.

☒ Life sciences      ☐ Behavioural & social sciences      ☐ Ecological, evolutionary & environmental sciences

For a reference copy of the document with all sections, see nature.com/documents/nr-reporting-summary-flat.pdf

# Life sciences study design

All studies must disclose on these points even when the disclosure is negative.

| | |
|---|---|
| Sample size | Raw data used in this study was collected by the ADES consortium, the ADSP consortium, the ADNI consortium, and researchers who collected the StEP-AD, Knight-ADRC, UCSF cohorts. Sample size was not pre-determined and was chosen based on all known available cohorts with relevant data collected to date, after quality control steps were performed in each cohort (described in detail in Supplementary Information) in particular to avoid any sample duplications and family relations. The sample size was calculated as the number of individuals summed across all studies in the meta-analysis, N=32,558. |
| Data exclusions | We excluded samples and variants from the analysis based on extensive quality control procedures as thoroughly explained in the Methods and Supplemental Note. Please see the Supplemental Note, Supplementary Tables 3, 4, and 5, on the effect on quality control measures taken (sample QC and variant QC). |
| Replication | After sample QC, we first compared gene-based rare-variant burdens between 12,652 AD cases and 8,693 controls in a Stage-1 analysis. To confirm burden signals identified in Stage 1, we applied an analysis model consistent with Stage-1 to an independent Stage-2 dataset, which after QC, comprised 3,384 cases and 7,829 controls. This confirmed the AD-association of rare damaging variants in the SORL1, TREM2, ABCA7, ATP8B4 and ABCA1 genes. The association signal of the ADAM10 gene was not exome-wide significant, presumably because the Stage-2 dataset encompassed too few prioritized variants in this gene. |
| Randomization | Data quality control was performed across all data, irrespective of case and control status. |
| Blinding | To perform burden analyses, the analysts were not blinded to the status of the individuals because this requires knowing case and control status. |

# Reporting for specific materials, systems and methods

We require information from authors about some types of materials, experimental systems and methods used in many studies. Here, indicate whether each material, system or method listed is relevant to your study. If you are not sure if a list item applies to your research, read the appropriate section before selecting a response.

## Materials & experimental systems

| n/a | Involved in the study |
|-----|----------------------|
| ☒ ☐ | Antibodies |
| ☒ ☐ | Eukaryotic cell lines |
| ☒ ☐ | Palaeontology and archaeology |
| ☒ ☐ | Animals and other organisms |
| ☐ ☒ | Human research participants |
| ☒ ☐ | Clinical data |
| ☒ ☐ | Dual use research of concern |

## Methods

| n/a | Involved in the study |
|-----|----------------------|
| ☒ ☐ | ChIP-seq |
| ☒ ☐ | Flow cytometry |
| ☒ ☐ | MRI-based neuroimaging |

# Human research participants

Policy information about studies involving human research participants

| | |
|---|---|
| Population characteristics | We used multiple independent sets of participants in this study. We adjusted the analysis for principal components. Sample sizes, age and gender characteristics for our sample can be found per cohort and in Supplementary Table 1 of the Supplementary Note, and in the cohort descriptions in Section 1.1 of the Supplementary Note. |
| Recruitment | Participants from case-control studies were primarily recruited from clinics, nursing homes, disease registries, and hospitals, with controls being drawn from various ongoing studies and screened to exclude dementia/cognitive decline (Please see recruitment procedures in the cohort descriptions in section 1.1 of the Supplementary Note). Cases were recruited according to clinical diagnosis and defined as probable AD cases with a potential risk of misdiagnosis (estimated between 10 and 20% in the literature). Controls included in the study were free of cognitive decline but a large part of them did not have any follow-up wit the possibility that they developed dementia years later. |
| Ethics oversight | Written informed consent was obtained from study participants or, for those with substantial cognitive impairment, from a caregiver, legal guardian, or other proxy, and the study protocols for all populations were reviewed and approved by the appropriate local Institutional Review Boards (see description of the samples in the Supplementary Note). |

Note that full information on the approval of the study protocol must also be provided in the manuscript.

