## [Peer Review File · Nature Genetics]

Peer Review Information

Manuscript Title: Exome sequencing identifies rare damaging variants in ATP8B4 and ABCA1 as risk factors for Alzheimer's Disease

Corresponding author name(s): Henne Holstege

Reviewer Comments & Decisions:

Decision Letter, initial version:

30th Sep 2021

Dear Dr Holstege,

Your Letter, "Exome sequencing identifies rare damaging variants in the ATP8B4 and ABCA1 genes as novel risk factors for Alzheimer's Disease." has now been seen by 2 referees. You will see from their comments below that while they find your work of interest, some important points are raised. We are interested in the possibility of publishing your study in Nature Genetics, but would like to consider your response to these concerns in the form of a revised manuscript before we make a final decision on publication.

We therefore invite you to revise your manuscript taking into account all reviewer comments. Please highlight all changes in the manuscript text file. At this stage we will need you to upload a copy of the manuscript in MS Word .docx or similar editable format.

*2) If you have not done so already please begin to revise your manuscript so that it conforms to our Letter format instructions, available

[here](http://www.nature.com/ng/authors/article_types/index.html).

*3) Include a revised version of any required Reporting Summary:

[REDACTED]

We hope to receive your revised manuscript within three to six months. If you cannot send it within this time, please let us know.

Nature Genetics is committed to improving transparency in authorship. As part of our efforts in this direction, we are now requesting that all authors identified as 'corresponding author' on published papers create and link their Open Researcher and Contributor Identifier (ORCID) with their account on the Manuscript Tracking System (MTS), prior to acceptance. ORCID helps the scientific community achieve unambiguous attribution of all scholarly contributions. You can create and link your ORCID from the home page of the MTS by clicking on 'Modify my Springer Nature account'. For more information please visit please visit

href="http://www.springernature.com/orcid">www.springernature.com/orcid.

Sincerely,

Wei

Wei Li, PhD
Senior Editor
Nature Genetics
New York, NY 10004, USA
www.nature.com/ng

Reviewers' Comments:

Reviewer #1:

Remarks to the Author:

This manuscript titled “Exome sequencing identifies rare damaging variants in the ATP8B4 and ABCA1 genes as novel risk factors for Alzheimer’s Disease” by Holstege, Hulsman, Charbonnier et al. (Nature Genetics MS# NG-LE58130) investigates the gene-based burden of rare damaging variants in exome sequencing (ES) data from 32,558 individuals, including 16,036 AD cases and 16,522 controls. These investigators essentially ‘harmonized and QC’d a lot of data’ and then used a two-stage analysis. Their data and analyses suggest that next to known genes TREM2, SORL1 and ABCA7 that there was a significant association of rare, predicted damaging (at least by REVEL, never saw how CADD or other ‘...flavor du jour did’), variants in ATP8B4 and ABCA1 with AD risk. They provide evidence for a suggestive association involving ADAM10 and SRC. Their analyses also highlighted RIN3, CLU, ZCWPW1 and ACE as potential “causal genes” in AD-GWAS loci. Thus, they propose rare damaging variants in these genes, and in particular loss-of-function (LoF) variants [null and hypomorphic, and perhaps antimorphic (dominant negative) alleles], have a large effect on AD-risk and they are enriched in early onset AD cases. They also suggest that the newly identified AD-associated genes provide additional

3evidence for a major role for APP-processing, A β -aggregation, microglial function in common disease consistent with the amyloid hypothesis for Alzheimer dementia.

The study represents a substantive effort, a huge amount of data, and analyses, and the results are likely to be of interest to the readers of Nature Genetics. Of particular interest was the 'linking' of common variant GWAS signals and rare variant alleles to the very same genes. There are, however, a number of things that could be clarified and might be helpful to the readers to further understand the impact of this important study.

1. It was not clear to this reviewer how, for the rare variants, variant confirmation was achieved? For the genes-of-interest were rare variant alleles Sanger confirmed? Was orthogonal ngs used? Did they at least do an independent 'wet bench' experimental confirmation of whatever genes/rare variant alleles, where their genomics discovery efforts identified rare variants in coding sequences to 'confirm reality' by an independent sequencing method such as Sanger dideoxynucleotide sequencing?

2. I can find no data in either the paper, or supplementary files, that relate to whether or not CNV analyses were performed in their studies. Such data AND analyses are essential given decades of literature that support the APP hypothesis wherein variants resulting from gene/genome copy number changes: Notably, Trisomy 21 in Down syndrome with an increased risk of early onset AD ((Delabar et al. (1987) β amyloid gene duplication in Alzheimer's disease and karyotypically normal Down syndrome. Science 235:1390-1392; Lupski et al. (2020) Clinical genomics and contextualizing genome variation in the diagnostic laboratory. Science 20:995-1002) and APP duplication CNV in Alzheimer Disease (Rovelet-Lecruix et al. (2006) APP locus duplication causes autosomal dominant early-onset Alzheimer disease with cerebral amyloid angiopathy. Nature Genetics 38:24-26).

3. This reviewer can't seem to find any experimental data to support these authors contention/speculation that GWAS SNPs reduce 'gene function'?

Minor edits:

4. I suggest a title change to : "Exome sequencing identifies rare damaging variants in ATP8B4 and ABCA1 as novel risk factors for Alzheimer Disease"

5. Page 6, paragraph 2 please italicize ABCA1 and CBX3. Moreover, I note that in several tables the gene symbols are not italicized and thus, represent a protein product encoded by the gene and not the gene itself.

6. I question the use of terminology/interpretation of 'causal genes' for disease as genes don't generate or cause disease. Variation of genes/genomes can lead to alterations in expression or gene function, that is gene action, and that may result in perturbations from biological homeostasis that affect downstream phenotypes which we characterize as disease and/or disease traits. It is not the gene itself that causes disease but variation in that gene function or regulation that contributes to the clinical manifestations we characterize as disease.

– Jim Lupski

Reviewer #2:

Remarks to the Author:

This paper is of interest to the general readership of Nature Genetics, increasing our knowledge of the genetics of Alzheimer's disease while also identifying 2 new potential therapeutic targets. Also, a massive effort relating to curating and cleaning such disparate and heterogenous exome data, impressive.

Some specific comments for the authors:

Power calcs would be helpful, exome studies in particular suffer from low power issues and many readers seem to not understand that 20K exomes is a bare minimum. Most readers don't understand that finding a couple genes using exome rare variant burden tests actually needs this level sample size and that is a major feat in and of itself. Maybe just a sentence to convey the scale of the paper.

Pedantic comment, apologies, but there is no such thing as "suggestive association in ADAM10 and SRC", it is either significant or not. Please remove reference to suggestive and/or marginal associations or reword these concepts.

Can drop the part about to $FDR < 0.2$. Raises stats reviewer red flags.

"The genes SORL1, TREM2, ABCA7, ATP8B4, ABCA1 and ADAM10 reached $p < 0.05$ while SRC reached $p = 0.07$ (Table 2A, Stage-2). Effect-sizes of these genes were concordant with those observed in Stage-1." Range of p-values and the r^2 or r for the effect estimates would be an improvement.

When looking at the rare variant burdens for GWAS hits, could this be explored also using MAGMA or similar method? If logistically complicated, no big deal for ignoring, but could potentially be interesting / confirmatory.

Be clear if these were adjusted for APOE4 or not. Would be of interest to compare APOE4 adjusted versus not adjusted for hits at least (if logistically possible).

"We found that the effect-sizes of rare, coding variant-burdens were large compared to the effect-sizes

5of the GWAS sentinel variants” this paragraph is not too clear and could likely be replaced. Might be easy to just show results of formally testing variant burdens per gene for association w/ age at onset. Also, “ must be associated with reduced activity of the gene” can be formally tested / backed up with stats instead of just stated ... how are the QTLs at these genes looking in AD?

“Genes were considered suggestively associated with AD in Stage-1 if the False Discovery Rate was <20% (FDR<0.2) (Benjamini-Hochberg procedure³²).” ← too soft of a significance threshold, it has gotta go. Why not just focus on both instead of the weighted mean, helps with interpretability for the average reader or a researcher attempting meta-analysis → “ordinal logistic regression can be interpreted as weighted averages of the OR of being an AD case versus control, and the OR of being an early-onset AD case or not”

For interpretability, generally provide FDR adjusted p-values instead of %.

UK Biobank summary stats for burdens may be helpful for this research (external validation), but may be logistically challenging to obtain comparable stats because of the variant annotation filtering ... this may be outside the scope of this paper.

There is no code or data access section. This would be really helpful, particularly as this is a strong example of how to deal with the “wild west” of exome analyses from very heterogenous sources, captures and contributing studies.

In the interest of fair, transparent and accountable reviews, feel free to reach out to me if there are any questions or if clarification is needed. This is a solid piece of research, just needs a little bit of tuning.

Good luck!

-

Mike A. Nalls, PhD

mike@datatecnica.com

Data science team lead - NIH's CARD and NIA's LNG

Founder / consultant – DTi

Data Tecnica International

@mike_nalls

he/him/his

Author Rebuttal to Initial comments

Please find attached our revised manuscript entitled: “*Exome sequencing identifies rare damaging variants in the ATP8B4 and ABCA1 genes as novel risk factors for Alzheimer’s Disease*”. We very much appreciated the positive response and thoughtful comments provided by the reviewers. We feel that the implementation of these comments has substantially improved our work. Please find a point-by-point response to each comment below.

Note: the page numbers and lines mentioned in the responses below correspond to page numbers and lines in the revised manuscript.

Reviewer #1:

Remarks to the Author: This manuscript titled “*Exome sequencing identifies rare damaging variants in the ATP8B4 and ABCA1 genes as novel risk factors for Alzheimer’s Disease*” by Holstege, Hulsman, Charbonnier et al. (Nature Genetics MS# NG-LE58130) investigates the gene-based burden of rare damaging variants in exome sequencing (ES) data from 32,558 individuals, including 16,036 AD cases and 16,522 controls. These investigators essentially ‘harmonized and QC’d a lot of data’ and then used a two-stage analysis. Their data and analyses suggest that next to known genes *TREM2*, *SORL1* and *ABCA7* that there was a significant association of rare, predicted damaging (at least by REVEL, never saw how CADD or other ‘...flavor du jour did’), variants in *ATP8B4* and *ABCA1* with AD risk. They provide evidence for a suggestive association involving *ADAM10* and *SRC*. Their analyses also highlighted *RIN3*, *CLU*, *ZCWPW1* and *ACE* as potential “causal genes” in AD-GWAS loci. Thus, they propose rare damaging variants in these genes, and in particular loss-of-function (LoF) variants [null and hypomorphic, and perhaps antimorphic (dominant negative) alleles], have a large effect on AD-risk and they are enriched in early onset AD cases. They also suggest that the newly identified AD-associated genes provide additional evidence for a major role for APP-processing, A β -aggregation, microglial function in common disease consistent with the amyloid hypothesis for Alzheimer dementia.

The study represents a substantive effort, a huge amount of data, and analyses, and the results are likely to be of interest to the readers of Nature Genetics. Of particular interest was the ‘linking’ of common variant GWAS signals and rare variant alleles to the very same genes. There are, however, a number of things that could be clarified and might be helpful to the readers to further understand the impact of this important study.

We want to thank the reviewer for his thoughtful review, in which he points out crucial strengths and weaknesses of our work. We have carefully considered and implemented all points raised by the reviewer, as indicated in a point-by-point discussion below. We feel that this has led to a greatly improved version of our manuscript.

To comment on the “flavour du jour” (REVEL score): since we are investigating coding sequences only, we chose to use a variant effect prediction algorithm that was trained on

coding sequence only, such as REVEL. The CADD score is trained on the entire genome, such that CADD may not be specialized in the specific idiosyncrasies pertaining to coding sequences.

Reviewer #1 Comment 1.

It was not clear to this reviewer how, for the rare variants, variant confirmation was achieved? For the genes-of-interest were rare variant alleles Sanger confirmed? Was orthogonal ngs used? Did they at least do an independent 'wet bench' experimental confirmation of whatever genes/rare variant alleles, where their genomics discovery efforts identified rare variants in coding sequences to 'confirm reality' by an independent sequencing method such as Sanger dideoxynucleotide sequencing?

Given the large number of identified rare variants it was logistically impossible to confirm each variant call with Sanger sequencing. To attain confidence that these calls are indeed true positives, we took the following measures:

- 1) We used posterior probabilities for variant calls: this method requires increased evidence (i.e., read coverage etc), for calls with (very) low population frequency, thereby reducing false positive calls.
- 2) We QC'ed to achieve an inflation close to 1.0, to preclude that false positive calls are biased towards cases or controls.
- 3) We estimated the number of novel variants (i.e., those unknown to dbSNP), for each sample. Then, we excluded samples for which the number of novel variants deviated significantly from those across the rest of the sample.
- 4) Additionally, a large number of other possible measures were considered that allowed us to detect possible biases related to false positive variant calls, as described in Supplement section 1.4, 1.6, 1.8.
- 5) To further improve the confidence in these statistical methods, we performed a validation step using an existing dataset containing Sanger validation calls for variants in the *SORL1* gene, the gene in which we detected by far the most variants.
 - a. In a subset of 1,908 samples (from the ADC and Rotterdam Study datasets), we detected 76 singleton variants, and (irrespective of QC status) we tested them all using Sanger sequencing¹. For the current work, we reanalyzed this dataset in the context of the current pipeline: of the 76 detected *SORL1* variants, N=41 *SORL1* variant calls passed QC in our current dataset and these were all confirmed through Sanger sequencing (**100% true positive rate**). For the remaining 35 *SORL1* variants: N=8 variants were not present in the current dataset due to sample exclusion (all flagged due to ≤ 3 rd degree family relations (IBD)). N=15 *SORL1* variants were excluded in the case-control analysis, as they were flagged by the QC as susceptible to batch issues (either due to differences in missingness between cases and controls, or flagged by the variant batch detector). For such variants, individual variant calls are usually still reliable, as batch effects are generally derived from the missing calls. Indeed, they were all confirmed through Sanger sequencing. Finally, N=14 *SORL1* variant calls were flagged/not called by our pipeline, and indeed were not confirmed with Sanger sequencing (**100% true negative rate**).

- b. We also obtained Sanger sequencing results for the Rouen study, where Sanger sequencing is performed as part of standard clinical practice and was also collected for several studies^{2,3}, some of which are not yet published. A total of 69 variant calls that passed QC were tested through Sanger sequencing: 28 in *SORL1*, 32 in *ABCA7* and 9 in *TREM2*. All variant calls were confirmed as true positives (**100% true negative rate**).

We have added these results in **section 1.11** in the supplement.

Reviewer #1 Comment 2.

I can find no data in either the paper, or supplementary files, that relate to whether or not CNV analyses were performed in their studies. Such data AND analyses are essential given decades of literature that support the *APP* hypothesis wherein variants resulting from gene/genome copy number changes: Notably, Trisomy 21 in Down syndrome with an increased risk of early onset AD ((Delabar et al. (1987) β amyloid gene duplication in Alzheimer's disease and karyotypically normal Down syndrome. *Science* 235:1390-1392; Lupski et al. (2020) Clinical genomics and contextualizing genome variation in the diagnostic laboratory. *Science* 20:995-1002) and *APP* duplication CNV in Alzheimer Disease (Rovelet-Lecruix et al. (2006) *APP* locus duplication causes autosomal dominant early-onset Alzheimer disease with cerebral amyloid angiopathy. *Nature Genetics* 38:24-26).

We agree with the reviewer about the importance of CNVs regarding Alzheimer disease risk. Until now, the most important CNV associated with AD is undoubtedly the duplication of the *APP* locus. In this work, we excluded known *APP* duplication carriers (in line with our exclusion of pathogenic variants carriers in Mendelian genes). Given prior results from the group of Rouen (Le Guennec et al., *Mol Psych* 2017 and unpublished work on a larger dataset), we do not expect that moderately or highly recurrent CNVs with a large effect exist, even among EOAD patients. Therefore, to identify rare CNVs, or common CNVs with smaller effects, a very large dataset is required. The current dataset will be likely well-suited for that purpose (or, maybe, even underpowered given the extreme rarity of most CNVs).

However, identification of CNVs will require additional strong methodological efforts. Beyond limitations due to statistical power, CNV detection is subject to specific and important methodological challenges, in addition to those we already faced for SNVs/indels. The current dataset has been obtained through heterogeneous sequencing techniques: both whole exome sequencing (with several different capture kits, read depth and fragment length) and whole genome sequencing. This prevents CNV calling using a harmonized pipeline. CNV callers using WGS data rely on read depth comparison, split reads, and/or pair orientation, while those adapted to WES data rely on read depth comparison with inter-individual normalization. These methods are highly sensitive to data heterogeneity and batch effects. Thus, extensive normalization is necessary to deal with data heterogeneity among WES samples on one side, and additional efforts are necessary to combine WES and WGS, with the aim to perform an unbiased case-control analysis. This effort is currently taking place within the consortium.

To indicate that future analyses should include structural variations and CNVs we added to the discussion on **page 9, line 22** on “[...] Further, the effect of structural variants such as CNVs and repetitive sequences will need to be investigated in future analyses.”

Reviewer #1 Comment 3.

This reviewer can't seem to find any experimental data to support these authors contention/speculation that GWAS SNPs reduce 'gene function'?

The objective of detecting associations of rare variants with AD risk in genes located within GWAS loci was threefold:

- (i) To prioritize the gene likely responsible for the GWAS signal in loci encompassing several genes;
- (ii) To highlight the importance of a GWAS gene in the pathophysiological process of AD;
- (iii) To propose hypotheses regarding the nature of the involvement of the gene in this AD etiology.

For the latter, we observed that of the variants observed in the prioritized GWAS genes, a substantial fraction of the rare variants that associated with increased AD risk were LOF variants. This suggests that a lower expression level of these genes is associated with increased AD risk. This, in turn, led us to speculate that risk elements in linkage with the common risk-increasing GWAS alleles (which are generally non-coding) will similarly impact the driver gene by reducing gene expression. We agree that this is a deductive reasoning, which at this point we cannot support with functional experimental evidence. However, we think that our data contribute to the further interpretation of the nature of the AD-risk identified in GWAS. Nevertheless, we fully agree with the reviewer that the effect of carrying a GWAS variant on the expression level of that gene will have to be further evaluated. To clarify this in the text we adjusted it as follows:

page 9, line 17:

“Given the association of LOF variants with increased AD-risk, we suggest that the GWAS risk alleles in the respective loci might also be associated with reduced activity of the gene, which will have to be evaluated in further experiments.”

Minor edits:

Reviewer #1 Comment 4. I suggest a title change to : “Exome sequencing identifies rare damaging variants in *ATP8B4* and *ABCA1* as novel risk factors for Alzheimer Disease”

We thank the reviewer for this suggestion, and have made this change.

Reviewer #1 Comment 5. Page 6, paragraph 2 please italicize *ABCA1* and *CBX3*. Moreover, I note that in several tables the gene symbols are not italicized and thus, represent a protein product encoded by the gene and not the gene itself.

We have changed this accordingly.

Reviewer #1 Comment 6. I question the use of terminology/interpretation of 'causal genes' for disease as genes don't generate or cause disease. Variation of genes/genomes can lead to alterations in expression or gene function, that is gene action, and that may result in perturbations from biological homeostasis that affect downstream phenotypes which we characterize as disease and/or disease traits. It is not the gene itself that causes disease but variation in that gene function or regulation that contributes to the clinical manifestations we characterize as disease.

We agree with the reviewer and have changed all mentions of the words **causal** or **culprit** into **driver**. As an example: the sentence (**page 5 line 7**):

"Next to these genes, our analysis highlighted RIN3, CLU, ZCWPW1 and ACE as potential causalgenes in AD-GWAS loci." was changed into: *"Next to these genes, the rare variant burden in RIN3, CLU, ZCWPW1 and ACE highlighted these genes as potential driver genes in AD-GWAS loci"*

– Jim Lupski

Reviewer #2:

Remarks to the Author: This paper is of interest to the general readership of Nature Genetics, increasing our knowledge of the genetics of Alzheimer's disease while also identifying 2 new potential therapeutic targets. Also, a massive effort relating to curating and cleaning such disparate and heterogenous exome data, impressive. Some specific comments for the authors:

Reviewer #2 Comment 1. Power calcs would be helpful, exome studies in particular suffer from low power issues and many readers seem to not understand that 20K exomes is a bare minimum. Most readers don't understand that finding a couple genes using exome rare variant burden tests actually needs this level sample size and that is a major feat in and of itself. Maybe just a sentence to convey the scale of the paper.

We thank the reviewer for this suggestion, and have added power analyses as a new **Figure 1B**, and an additional **Table S4**. As suggested by the reviewer, in the current discovery dataset we have limited power to attain $p < 1e-6$, even for burdens with a large OR. For example: we only have 22% power to identify an OR of 10.0 in EOAD cases (or an OR of 3.33 in LOAD cases) for the LOF+REVEL \geq 50 category. This indicates the necessity for a substantial increase in the sample size to identify the gene burdens of variants with a major impact on AD.

To reflect this, we added the following sentence to the manuscript (**page 6, line 8**):

*"Of the 19,822 autosomal protein coding genes, we analyzed the 13,222 genes that had a cumulative minor allele count (cMAC) \geq 10 for the lowest deleterious threshold LOF+REVEL \geq 25 (see **Methods**); 9,168 genes for the LOF+REVEL \geq 50 threshold; 5,694 for the LOF+REVEL \geq 75 threshold and 3,120 genes for the LOF-only threshold (**Figure 1B**). For these different*

deleteriousness thresholds, this analysis has an estimated power of 41%, 22%, 11% and 4%, respectively to attain a signal with $p < 1e-6$, assuming that the differential variant burden for a gene is associated with an odds ratio of 10.0 in EOAD and 3.33 in LOAD (Table S4). Therefore, this analysis only has the power to uncover genes for which the differential gene-burden is associated with a large effect size or large numbers of damaging variant carriers (Figure 1B)."

Reviewer #2 Comment 2. Pedantic comment, apologies, but there is no such thing as "suggestive association in ADAM10 and SRC", it is either significant or not. Please remove reference to suggestive and/or marginal associations or reword these concepts.

We agree with the reviewer and changed all mentions of "suggestive association" to 'suggestive signal' and we now state more clearly that no significant association was reached for ADAM10. Note that we removed all mentions of the 'suggestive' signal in the SRC gene we identified in the Stage 1 analysis, following the change to a more stringent FDR cutoff < 0.1 (see our answer to comment 3 below).

Changes include (**page 7 line 4**):

Old text: *"While our data support the AD-association in the ADAM10 gene, variants in this gene are extremely few and rare, such that the signal can only be confirmed in larger datasets."*

New text: *"The association signal of the ADAM10 gene was not exome-wide significant, presumably because prioritized variants in this gene are extremely few and rare, such that the signal can be confirmed only in larger datasets."*

Reviewer #2 Comment 3. Drop the part about to FDR < 0.2. Raises stats reviewer red flags.

While we only declare signals significant once they pass a strict Holm-Bonferroni test, we understand the concern of this reviewer that using an FDR cutoff of < 0.2 in Stage-1 may still raise a 'red flag'. In the current version of the manuscript, we have lowered the threshold to FDR < 0.1 , which is commonly used in discovery stages, for example in two recent large exome sequencing studies, one by Satterstrom et al., on autism disorders⁴ and one by Tian et al, on depressive disorders using the UK Biobank exome sequencing study⁵. Following from this adaptation, we removed the signals in the MTO1, CBX3, PRSS3, B3GNT4 and SRC genes from the text, tables, and figures.

Reviewer #2 Comment 4. "The genes SORL1, TREM2, ABCA7, ATP8B4, ABCA1 and ADAM10 reached $p < 0.05$ while SRC reached $p = 0.07$ (Table 2A, Stage-2). Effect-sizes of these genes were concordant with those observed in Stage-1." Range of p-values and the r^2 or r for the effect estimates would be an improvement.

We have removed SRC from this sentence (see comment 3), and have now added the correlation between effect sizes to the text. We refer to Table 2A for the p-value per gene. The sentence now reads (**page 6 line 24**): *"All genes selected in Stage-1 reached $p < 0.05$ (Table 2A,*

Stage-2). Stage-2 effect-sizes of these genes correlated with those observed in Stage-1 (Pearson's r on log-odds: 0.91)."

Reviewer #2 Comment 5. When looking at the rare variant burdens for GWAS hits, could this be explored also using MAGMA or similar method? If logistically complicated, no big deal for ignoring, but could potentially be interesting / confirmatory.

We have considered these analyses. However, as revealed in our results, the signal is mainly derived from extremely rare variants, which cannot be imputed in the GWAS signal. Any signal therefore uncovered by MAGMA or similar methods in GWAS data would rely only on more common variants, and therefore would be derived from an independent signal. Such signals, if uncovered, would also be difficult to assign to a specific gene, due to linkage between common variants. Moreover, in our GWAS loci-focused analysis, we used results from Schwarzenuber et al. to prioritize genes. Schwarzenuber et al. rely on conditional analyses and fine-mapping to try to overcome the aforementioned issue. As such, adding a confirmatory analysis to the burden test which relies on GWAS data would resemble a circular argument. We believe therefore that MAGMA or related analyses better fit the scope of a GWAS study.

Reviewer #2 Comment 6. Be clear if these were adjusted for APOE4 or not. Would be of interest to compare APOE4 adjusted versus not adjusted for hits at least (if logistically possible).

Samples from some of the contributing datasets (including the large ADSP dataset), were a priori selected based upon APOE status. Therefore, any correction for APOE genotype in our Stage 1 analysis would bias the analyses towards features in these cohorts. Due to this, we have chosen not to correct for APOE in our main analysis.

To make this clearer we write in the manuscript (**page 9, line 8**):

"Since APOE status was used as selection criterion in several contributing datasets, burden tests were not adjusted for APOE- ϵ 4 dosage; in a separate analysis we observed no interaction-effects between the rare-variant AD-association and APOE- ϵ 4 dosage (Table S11, Online Methods)."

Reviewer #2 Comment 7. "We found that the effect-sizes of rare, coding variant-burdens were large compared to the effect-sizes of the GWAS sentinel variants" this paragraph is not too clear and could likely be replaced. Might be easy to just show results of formally testing variant burdens per gene for association w/ age at onset.

We took the comment of the reviewer to heart and changed the text to accommodate additional testing results. (**page 8, line 15**):

"For damaging variants in most genes, we observed increased carrier frequencies in younger cases and larger effect sizes were associated with an earlier age at onset ($p=0.0001$) (Table S7)."

We also added results to the paragraph on **page 8, line 23**:

“Extremely rare variants contributed more to large effect sizes than less rare variants ($p=0.03$, Table S8).

Finally, we adapted the paragraph in the discussion to which the reviewer refers (**page 9, line 17**):

“Given the association of LOF variants with increased AD-risk, we suggest that the GWAS risk alleles in the respective loci might also be associated with reduced activity of the gene, which will have to be evaluated in further experiments. We observed an increased burden of rare damaging genetic variants in individuals with an earlier age at onset. Nevertheless, damaging variants (including APOE- $\epsilon 4/\epsilon 4$) were observed in only 30% of the EOAD cases (Table S10), suggesting that additional damaging variants remain to be discovered (Figure 1B). Further, the effect of structural variants such as CNVs and repetitive sequences will need to be investigated in future analyses.”

Reviewer #2 Comment 8. Also, “must be associated with reduced activity of the gene” can be formally tested / backed up with stats instead of just stated ... how are the QTLs at these genes looking in AD?

For this statement, we applied deductive reasoning based on the assumption that LOF variants present the strongest biological evidence for reduced protein activity. (Note that this question was also raised by Reviewer 1 in comment 1.3.)

We agree with the reviewer that eQTL/sQTLs analyses may potentially support our reasoning, but at current, there are several limitations restricting the pertinence of such data and subsequent TWASs: the publicly available databases based on AD brain samples are still limited such that effects associated with disease stage, age at onset, disease heterogeneity, and other features cannot be taken into consideration. In addition, eQTLs can differ per cell-type, representing an additional level of complexity.

To illustrate these issues, we focused on *SORL1*, which was previously shown to be haploinsufficient: decreased *SORL1* expression (i.e., due to a LOF variant) is associated with an increased risk of AD, dysregulated APP metabolism, increased Ab secretion as reviewed by Champion et al.⁶ Also, *SORL1* risk haplotypes have been associated with a suggestive reduced expression in response to BDNF in induced neurons⁷. We set out to assess eQTLs of the GWAS SNPs in this gene by interrogating the eQTL catalog (fivex.sph.umich.edu) which presents eQTL data from numerous studies. We filtered results based on the prioritized gene and brain tissues, after setting up a significance threshold of $-\log_{10}(p) = 2.00$. The *SORL1* risk allele appeared to be associated with *increased* expression, which is in apparent contradiction with previous findings. Together, this indicates that QTLs currently do not fully capture the functionality of variants and their potential impact on gene expressions in a complex pathological context.

Therefore, to take into account the relevant comment of the reviewer, we have now more carefully worded the highlighted sentence as follows (**page 9 line 17**): *“Given the association of LOF variants with increased AD-risk, we suggest that the GWAS risk alleles in the respective loci might also be associated with reduced activity of the gene, which will have to be evaluated in further experiments.”*

Reviewer #2 Comment 9. “Genes were considered suggestively associated with AD in Stage-1 if the False Discovery Rate was <20% (FDR<0.2) (Benjamini-Hochberg procedure32).” ← too soft of a significance threshold, it has gotta go.

We have reduced this threshold to FDR<0.1 (see point 3).

Reviewer #2 Comment 10. Why not just focus on both instead of the weighted mean, helps with interpretability for the average reader or a researcher attempting meta-analysis → “ordinal logistic regression can be interpreted as weighted averages of the OR of being an AD case versus control, and the OR of being an early-onset AD case or not”

As the reviewer pointed out, exome analyses are restricted in available power. Performing two separate, but overlapping tests would further reduce power due to the additional required multiple testing correction. Moreover, the signal will usually be neither exclusively EOAD < LOAD ~ CONTROL (domain of very early onset familial genes like *APP*, *PSEN1*) nor EOAD ~ LOAD < Control (domain of GWAS hits with small effect sizes). Rather, in an exome analysis in which we aim to identify novel AD-associated genes in which variants can have a large impact, we will often end up between the two extremes, i.e.: EOAD < LOAD < Control. A test that is geared specifically to identify signals in this setting is therefore most promising. To support this, we added a power analysis (**Figure 1B and Table S4**). To illustrate this (referring also to Comment 1), for the hypothetical setting in which EOAD OR = 10.0, and LOAD OR = 3.33, and for a gene with 25 damaging allele carriers, a case/control test would have 4% power to observe a signal at $p < 1e-6$ significance level; a test which compares EOAD to the other samples (EOAD < LOAD ~ Control) has 22% power, while the ordinal test (EOAD < LOAD < Control) has 35% power. Note that these power estimates do not yet take into account the additional multiple testing correction needed when performing *both* a case/control and EOAD vs. rest test. Note that for interpretation purposes, we fully agree with the reviewer that separate EOAD and LOAD effect size estimates are advantageous, and the appropriate ORs are therefore included in **Table 3**.

Reviewer #2 Comment 11. For interpretability, generally provide FDR adjusted p-values instead of %s.

We have adjusted all FDR cutoffs to fractional values.

Reviewer #2 Comment 12. UK Biobank summary stats for burdens may be helpful for this research (external validation), but may be logistically challenging to obtain

comparable stats because of the variant annotation filtering ... this may be outside the scope of this paper.

We plan to make use of this rich resource in follow-up work. As indicated by the reviewer, this is a large undertaking and our focus on early onset AD might not translate well to this dataset, as we would have to carefully consider how to handle the (proxy) phenotypes. Given the low AD incidence at young ages, the dataset is likely to function predominantly as an extra control dataset, and not as a dataset for independent replication of early-onset-focused signals

Reviewer #2 Comment 13. There is no code or data access section. This would be really helpful, particularly as this is a strong example of how to deal with the “wild west” of exome analyses from very heterogenous sources, captures and contributing studies.

During the time of writing this rebuttal, code has been made available on holstegelab.eu/tools, which guides the reader to https://github.com/holstegelab/seq_qc. Furthermore, analyses strategies are described in detail in the supplement for each step in the process.

Moreover, summary statistics of the discovery analysis will be made available on holstegelab.eu/data upon the acceptance of this manuscript. For all tests with a cMAC ≥ 10 , this will include Ensembl gene id, gene name, variant category, cMAC, pvalue, beta, se.

Reviewer #2 Comment 14. In the interest of fair, transparent and accountable reviews, feel free to reach out to me if there are any questions or if clarification is needed. This is a solid piece of research, just needs a little bit of tuning. Good luck!

Mike A. Nalls, PhD

mike@datatecnica.com

Data science team lead - NIH's CARD and NIA's LNG

Founder / consultant – DTi

Data Tecnica International

@mike_nalls

he/him/his

1. Holstege, H. *et al.* Characterization of pathogenic SORL1 genetic variants for association with Alzheimer's disease: a clinical interpretation strategy. *Eur J Hum Genet* **25**, 973-981 (2017).
2. Nicolas, G. *et al.* SORL1 rare variants: a major risk factor for familial early-onset Alzheimer's disease. *Mol Psychiatry* (2015).
3. Bellenguez, C. *et al.* Contribution to Alzheimer's disease risk of rare variants in TREM2, SORL1, and ABCA7 in 1779 cases and 1273 controls. *Neurobiol Aging* **59**, 220 e1-220 e9 (2017).
4. Satterstrom, F.K. *et al.* Large-Scale Exome Sequencing Study Implicates Both Developmental and Functional Changes in the Neurobiology of Autism. *Cell* **180**, 568-584.e23 (2020).

5. Tian, R. *et al.* Whole exome sequencing in the UK Biobank reveals risk gene SLC2A1 and biological insights for major depressive disorder. *MedRxiv* (2021).
6. Champion, D., Charbonnier, C. & Nicolas, G. SORL1 genetic variants and Alzheimer disease risk: a literature review and meta-analysis of sequencing data. *Acta Neuropathol* (2019).
7. Young, J.E. *et al.* Elucidating molecular phenotypes caused by the SORL1 Alzheimer's disease genetic risk factor using human induced pluripotent stem cells. *Cell Stem Cell* **16**, 373-85 (2015).

Decision Letter, first revision:

Our ref: NG-LE58130R

18th Mar 2022

Dear Dr. Holstege,

Thank you for submitting your revised manuscript "Exome sequencing identifies rare damaging variants in ATP8B4 and ABCA1 as novel risk factors for Alzheimer's Disease" (NG-LE58130R). It has now been seen by the original referees and their comments are below. The reviewers find that the paper has improved in revision, and therefore we'll be happy in principle to publish it in Nature Genetics, pending minor revisions to comply with our editorial and formatting guidelines.

Sincerely,
Wei

Wei Li, PhD
Senior Editor
Nature Genetics
New York, NY 10004, USA
www.nature.com/ng

7Reviewer #1 (Remarks to the Author):

Apologize profusely for my slight delay in the RE-REVIEW process.

I enjoyed the AUTHORS Pt2Pt RESPONSES to my and OTHER REVIEWERS COMMENTS/queries.

Nice paper, I look forward to reading again, and IN-DEPTH, when it appears.

- Jim Lupski

Reviewer #2 (Remarks to the Author):

All queries addressed. Thanks for the effort. Nice work.

-

Mike A. Nalls, PhD

Project director, data science - NIH's CARD and NIA's LNG

Founder / consultant – DTi

Data Tecnica International

@mike_nalls

he/him/his

Author Rebuttal to first revision comments

Point-by-point response to the reviewers' comments

Please find attached our revised manuscript entitled: "*Exome sequencing identifies rare damaging variants in the ATP8B4 and ABCA1 genes as novel risk factors for Alzheimer's Disease*". We very much appreciated the positive response and thoughtful comments provided by the reviewers during their first round of reviews. We feel that the implementation of these comments has substantially improved our work. The reviewers had no comments on our revised manuscript.

Reviewer #1:

Remarks to the Author: Apologize profusely for my slight delay in the RE-REVIEW process. I enjoyed the AUTHORS Pt2Pt RESPONSES to my and OTHER REVIEWERS COMMENTS/queries. Nice paper, I look forward to reading again, and IN-DEPTH, when it appears. - Jim Lupski

We greatly thank the reviewer for his time to review our manuscript, and for his nice comments.

Reviewer #2:

Remarks to the Author: All queries addressed. Thanks for the effort. Nice work. - Mike A. Nalls, PhD. Project director, data science - NIH's CARD and NIA's LNG, Founder / consultant – Dti Data Tecnica International, @mike_nalls; he/him/his

We greatly thank the reviewer for his time to review our manuscript, and for his nice comments.

Final Decision Letter:

In reply please quote: NG-LE58130R1 Holstege

19th Sep 2022

Dear Dr. Holstege,

I am delighted to say that your manuscript "Exome sequencing identifies rare damaging variants in ATP8B4 and ABCA1 as risk factors for Alzheimer's Disease" has been accepted for publication in an upcoming issue of Nature Genetics.

Your paper will be published online after we receive your corrections and will appear in print in the next available issue. You can find out your date of online publication by contacting the Nature Press Office (press@nature.com) after sending your e-proof corrections. Now is the time to inform your Public Relations or Press Office about your paper, as they might be interested in promoting its publication. This will allow them time to prepare an accurate and satisfactory press release. Include your manuscript tracking number (NG-LE58130R1) and the name of the journal, which they will need when they contact

9our Press Office.

Please note that *Nature Genetics* is a Transformative Journal (TJ). Authors may publish their research with us through the traditional subscription access route or make their paper immediately open access through payment of an article-processing charge (APC). Authors will not be required to make a final decision about access to their article until it has been accepted. [Find out more about Transformative Journals](https://www.springernature.com/gp/open-research/transformative-journals)

Authors may need to take specific actions to achieve [compliance](https://www.springernature.com/gp/open-research/funding/policy-compliance-faqs) with funder and institutional open access mandates. If your research is supported by a funder that requires immediate open access (e.g. according to [Plan S principles](https://www.springernature.com/gp/open-research/plan-s-compliance)) then you should select the gold OA route, and we will direct you to the compliant route where possible. For authors selecting the subscription publication route, the journal's standard licensing terms will need to be accepted, including [self-archiving-and-license-to-publish](https://www.nature.com/nature-portfolio/editorial-policies/self-archiving-and-license-to-publish). Those licensing terms will supersede any other terms that the author or any third party may assert apply to any version of the manuscript.

Please note that Nature Portfolio offers an immediate open access option only for papers that were first submitted after 1 January, 2021.

10If you have posted a preprint on any preprint server, please ensure that the preprint details are updated with a publication reference, including the DOI and a URL to the published version of the article on the journal website.

If you have not already done so, we invite you to upload the step-by-step protocols used in this manuscript to the Protocols Exchange, part of our on-line web resource, natureprotocols.com. If you complete the upload by the time you receive your manuscript proofs, we can insert links in your article that lead directly to the protocol details. Your protocol will be made freely available upon publication of your paper. By participating in natureprotocols.com, you are enabling researchers to more readily reproduce or adapt the methodology you use. [Natureprotocols.com](https://natureprotocols.com) is fully searchable, providing your protocols and paper with increased utility and visibility. Please submit your protocol to <https://protocolexchange.researchsquare.com/>. After entering your [nature.com](https://www.nature.com) username and password you will need to enter your manuscript number (NG-LE58130R1). Further information can be found at <https://www.nature.com/nature-portfolio/editorial-policies/reporting-standards#protocols>

Sincerely,
Wei

11Wei Li, PhD
Senior Editor
Nature Genetics
New York, NY 10004, USA
www.nature.com/ng